# Interplay between acetylation and ubiquitination of imitation switch chromatin remodeler Isw1 confers multidrug resistance in *Cryptococcus neoformans*

Yang Meng[1], Yue Ni[1], Zhuoran Li[1], Tianhang Jiang[1], Tianshu Sun[2], Yanjian Li[1], Xindi Gao[1], Hailong Li[3], Chenhao Suo[1], Chao Li[1], Sheng Yang[1], Tian Lan[1], Guojian Liao[4], Tongbao Liu[5], Ping Wang[6], Chen Ding[1]*

[1]College of Life and Health Sciences, Northeastern University, Shenyang, China; [2]Department of Scientific Research, Chinese Academy of Medical Sciences and Peking Union Medical College, Beijing, China; [3]NHC Key Laboratory of AIDS Immunology, The First Affiliated Hospital of China Medical University, Shenyang, China; [4]College of Pharmaceutical Sciences, Southwest University, Chongqing, China; [5]Medical Research Institute, Southwest University, Chongqing, China; [6]Department of Microbiology, Immunology, and Parasitology, Louisiana State University Health Sciences Center New Orleans, New Orleans, United States

*For correspondence: dingchen@mail.neu.edu.cn

**Abstract** *Cryptococcus neoformans* poses a threat to human health, but anticryptococcal therapy is hampered by the emergence of drug resistance, whose underlying mechanisms remain poorly understood. Herein, we discovered that Isw1, an imitation switch chromatin remodeling ATPase, functions as a master modulator of genes responsible for *in vivo* and *in vitro* multidrug resistance in *C. neoformans*. Cells with the disrupted *ISW1* gene exhibited profound resistance to multiple antifungal drugs. Mass spectrometry analysis revealed that Isw1 is both acetylated and ubiquitinated, suggesting that an interplay between these two modification events exists to govern Isw1 function. Mutagenesis studies of acetylation and ubiquitination sites revealed that the acetylation status of Isw1$^{K97}$ coordinates with its ubiquitination processes at Isw1$^{K113}$ and Isw1$^{K441}$ through modulating the interaction between Isw1 and Cdc4, an E3 ligase. Additionally, clinical isolates of *C. neoformans* overexpressing the degradation-resistant *ISW1*$^{K97Q}$ allele showed impaired drug-resistant phenotypes. Collectively, our studies revealed a sophisticated acetylation–Isw1–ubiquitination regulation axis that controls multidrug resistance in *C. neoformans*.

## Editor's evaluation

This important study makes a solid connection between chromatin remodeling, post-translational regulation, and antifungal drug resistance in Cryptococcus neoformans, revealing a new facet of how drug resistance can emerge. Establishing a link between chromatin remodeling and antifungal resistance is a finding that will be of interest to infectious disease researchers, cell biologists, and drug developers.

## Introduction

Emerging and re-emerging fungal pathogens are one of the primary causes of infectious diseases resulting in mortalities in humans and animals (*Brown et al., 2012*; *Fisher et al., 2012*; *Gnat et al., 2021*). It is estimated that approximately 300 million people suffer from fungal-related diseases, subsequently leading to 1.6 million deaths annually (*Stop neglecting fungi, 2017*). Additionally, fungal infection leads to substantial population declines in animals and amphibians, even threatening their extinction (*Fisher and Garner, 2020*; *Fisher et al., 2012*; *Scheele et al., 2019*). Intriguingly, certain pathogenic fungi are linked with cancer development (*Hosseini et al., 2022*; *Narunsky-Haziza et al., 2022*; *Saftien et al., 2023*). The World Health Organization has recently published the first-ever list of fungi posing threats to human health that lists *Cryptococcus neoformans* as one of four high-priority infection agents (*World Health Organization, 2022*). *Cryptococcus* species, which cause meningo-encephalitis and acute pulmonary infections, are responsible for 15% of the global HIV/AIDS-related fatalities (*Erwig and Gow, 2016*; *Kronstad et al., 2012*), totaling an estimated 220,000 deaths per year (*Iyer et al., 2021*; *Ngan et al., 2022*; *Rajasingham et al., 2017*; *Tugume et al., 2023*). Recently, it has been reported that *Cryptococcus* is also involved in secondary infections in people with COVID-19 (*Khatib et al., 2021*; *Spadari et al., 2020*; *Woldie et al., 2020*).

Cryptococcosis has a 100% death rate if patients are left untreated, but treatment remains challenging because the number of available antifungal medications is limited. Amphotericin B (Amp B) and azoles constitute the primary treatment options, with Amp B often in combination of 5-fluorocytosine (5-FC) (*Billmyre et al., 2020*; *Molloy et al., 2018*; *Spadari et al., 2020*). In light of the treatment limitations and risks, as well as the high costs associated with developing new antifungal treatments, the US FDA has categorized anti-cryptococcal therapies as 'orphan drugs', granting regulatory support by reducing the requirements for clinical studies (*Denning and Bromley, 2015*). Nevertheless, resistance to anti-cryptococcal drugs occurs rapidly, outpacing the development of new therapeutic options.

The mechanisms by which antifungal resistance occurs can be classified as either mutation in drug targets or epigenetic phenomena (*Lee et al., 2023*; *Li et al., 2020*; *Wan et al., 2021*). Point mutations in binding domains or regions of the drug target proteins often hinder their interactions with drugs resulting in resistance (*Sionov et al., 2012*; *Wan et al., 2021*), such as in the case of lanosterol demethylase, encoded by the *ERG11* gene, in *Candida* and *Cryptococcus* species (*Bosco-Borgeat et al., 2016*; *Sionov et al., 2012*). Additionally, pathogenic fungi display a disparate set of mutations that counteract various antifungal drugs. Defects in DNA mismatch repairs resulting in high gene mutation rates and, thereby, 5-FC resistance was identified in *C. deuterogattii,* a species distinct but close to *C. neoformans* (*Billmyre et al., 2020*). Mutations in genes encoding the cytosine permease (Fcy2), uracil phosphoribosyl transferase (Fur1), and UDP-glucuronic acid decarboxylase (Uxs1) conferring 5-FC resistance were also identified in multiple clinical cryptococcal isolates (*Billmyre et al., 2020*; *Chang et al., 2021*).

In comparison, nondrug target-induced resistance often refers to the association between mutations and altered gene expressions in regulatable components of sterol biosynthesis and efflux drug pumps (*Coste et al., 2004*; *Dunkel et al., 2008*; *Morschhäuser et al., 2007*; *Silver et al., 2004*). Studies of *C. albicans* transcription factor Tac1 illustrate that gain-of-function mutations or alterations in gene copy numbers regulate efflux pump genes (*Coste et al., 2004*). Gain-of-function mutations were also identified in *C. albicans* transcription factors Upc2 and Mrr1, which activate *ERG11* gene expression and drug pumps, respectively (*Dunkel et al., 2008*; *Silver et al., 2004*). A Upc2 homolog was identified in *C. neoformans* to be involved in regulating steroid biosynthesis, but its role in drug resistance was not clear (*Kim et al., 2010*). Other proteins capable of modulating efflux pump gene transcription in *Cryptococcus* remain unknown.

Still, a significant number of clinical isolates demonstrated resistance to multiple drugs, despite the absence of alterations in genes associated with drug resistance (*Malavia-Jones et al., 2023*; *Rajasingham et al., 2017*; *Sun et al., 2014*). Recent evidence indicates that epiregulation by protein posttranslational modification (PTM) mediates drug resistance in pathogenic fungi (*Calo et al., 2014*; *Robbins et al., 2012*). The acetylation of the heat-shock protein Hsp90 mediates antifungal drug response in *Saccharomyces cerevisiae* and *C. albicans* by blocking the interaction between Hsp90 and calcineurin (*Robbins et al., 2012*). Despite this observation, the functions of substantially acetylated proteins in antifungal drug resistance remain unknown (*Li et al., 2019*). Other important PTMs, such as ubiquitination, remain uninvestigated in human fungal pathogens.

Chromatin remodeling factors have significant roles in several physiological activities, such as gene transcription in response to environmental stressors. The ISWI (Imitation Switch) family is a well conserved group of chromatin remodeling families. The ISWI has a highly conserved ATPase domain from the SWI2/SNF2 family, which belongs to the superfamily of DEAD/H-helicases. This domain is responsible for driving chromatin remodeling. Additionally, the protein has distinctive HAND-SANT-SLIDE domains that are involved in binding to DNA. The presence of ISWI ATPase in chromatin remodeling complexes, including as NURF, CHRAC, and ACF, was first discovered in *Drosophila* homologs (*Tyagi et al., 2016*). Subsequently, it was found that these complexes are widely conserved in various other taxa, including yeast and mammals. ISWI complexes from different taxa have variations in their subunits. In *Drosophila*, there is a single ISWI gene that forms three separate complexes known as NURF, CHRAC, and ACF (*Längst and Becker, 2001*). The ISWI–ACF complex in *Neurospora crassa* is essential for repressing a certain set of genes that have been methylated at the H3K27, as well as genes targeted by PRC2 (polycomb repressed complex 2) (*Kamei et al., 2021*; *Wiles et al., 2022*). The baker's yeast contains two similar versions of Isws, known as Isw1 and Isw2. Isw1 forms associations with either Ioc3 or Ioc2 and Ioc4 to create two separate remodeling complexes, known as Isw1a and Isw1b, respectively (*Vary et al., 2003*). The *S. cerevisiae* Isw2 protein has the capability to interact with Itc1, forming a complex that can position the remodeler during the process of nucleosome sliding (*Kagalwala et al., 2004*). The ISWI mechanism in *S. cerevisiae* functions with assistance from other modulators. For instance, the Isw1b complex acts in conjugation with Chd1 to regulate chromatin organization (*Smolle et al., 2012*), and the ability of Isw2 to attach to DNA might be dependent on Sua7 (*Yadon et al., 2013*).

Nevertheless, there is currently no information available regarding ISWI in *C. neoformans*. In this study, we identified a conserved chromatin remodeler, Isw1, from *C. neoformans*, and we demonstrated its critical function in modulating gene expression responsible for multidrug resistance, as the *isw1* null mutant is resistant to azoles, 5-FC, and 5-fluorouracil (5-FU). We further demonstrated that Isw1 typifies a PTM interplay between acetylation and ubiquitination in regulating an Isw1 ubiquitin-mediated proteasome axis in response to antifungal exposure. Dissection of PTM sites on Isw1 revealed an essential reciprocal function of Isw1$^{K97}$ acetylation in modulating Isw1's binding to an E3 ligase, Cdc4, which initiates a ubiquitination–proteasome degradation process. Finally, we showed that the PTM interplay mechanism occurs horizontally in clinical strains of *C. neoformans*.

## Results

### Cryptococcal Isw1 represses drug resistance

Our previous studies demonstrated the indispensable function of acetylation in modulating the pathogenicity of *C. neoformans* (*Li et al., 2019*). Through screening our in-house acetylome knockout library, we found that a single homolog of imitation switch (ISWI)-class ATPase subunit, Isw1, is a critical modulator of multidrug resistance in *C. neoformans*. This is in contrast to *S. cerevisiae* in which whole-genome duplication led to the presence of two paralogs, Isw1 and Isw2 (*Kellis et al., 2004*; *Tsukiyama et al., 1999*; *Wolfe and Shields, 1997*). We have generated the cryptococcal *isw1Δ* mutant and complemented the mutant with the wild-type *ISW1* tagged with a Flag sequence (*Supplementary file 1*). The *isw1Δ* mutant exhibited profound resistance to azole compounds, including fluconazole (FLC), ketoconazole (KTC), 5-FC, and 5-FU, but not to the polyene compound Amp B (*Figure 1a*), in comparison to the wild-type and the complemented strains. Both liquid growth and agar spotting assays consistently showed robust resistance to all four antifungal compounds by the *isw1Δ* mutant strain (*Figure 1b*).

To examine whether Isw1-regulated multidrug resistance occurs at the transcription level of *ISW1* or its target genes, qRT-PCR of *ISW1* and transcriptome analysis of the wild-type and *isw1Δ* mutant strains treated with or without FLC was performed (*Supplementary file 2*). While the gene expression of *ISW1* remains unchanged in the presence of drugs (*Figure 1—figure supplement 1a*), the expression of genes important in drug resistance (*Denning and Bromley, 2015*), including seven genes encoding ATP-binding cassette (ABC) transporters (*CNAG_02296*, *CNAG_02430*, *CNAG_06338*, *CNAG_00792*, *CNAG_01575*, *CNAG_03600*, and *CNAG_02764*), two genes encoding efflux proteins (*CNAG_03101* and *CNAG_01960*), and three genes encoding major facilitator superfamily (MFS) transporters (*CNAG_01674*, *CNAG_04898*, and *CNAG_06259*), was significantly elevated in expression in

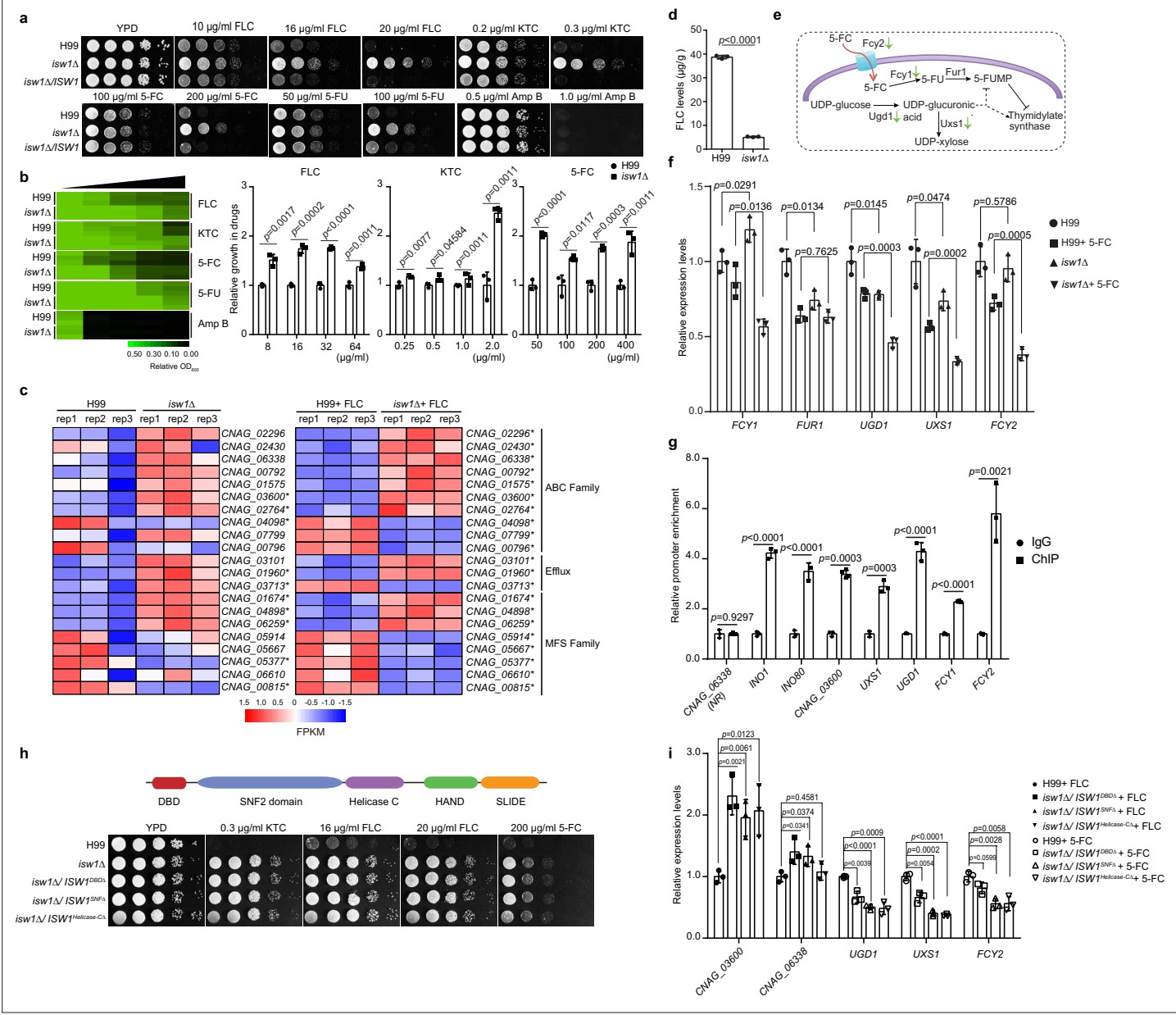

**Figure 1.** Isw1 represses the expression of drug-resistance genes. (**a**) Spotting assays of ISW1 mutant strains. Wild-type H99, *isw1Δ*, and ISW1 complementation strains were spotted onto YPD agar supplemented with indicated concentrations of antifungal agents. Plates were incubated at 30°C for 3 days. (**b**) Drug inhibitory tests. The H99 and *isw1Δ* (*n* = 3 each) strains were tested to determine the drug inhibition of several antifungal agents. Twofold dilutions of fluconazole (FLC) from 0 to 64 µg/ml, ketoconazole (KTC) from 0 to 2 µg/ml, or 5-fluorocytosine (5-FC) from 0 to 400 µg/ml were added to YPD medium. After 24 hr at 30°C, absorbance at 600 nm was used to measure growth. The optical densities of duplicate measurements were averaged and normalized relative to the control without FLC. Quantitative data were depicted in color (see color bar) and bar plots. Two-tailed unpaired *t*-tests were used. Data are expressed as mean ± standard deviation (SD). (**c**) Transcriptome analysis of *isw1Δ*. Samples of RNA were isolated from H99 and *isw1Δ* cells (*n* = 3 each) supplemented with or without 10 µg/ml FLC. Transcriptome analysis was performed, and a heat map of the expressions of drug-resistance genes was generated. Statistical significant genes are labeled with asterisks. (**d**) Intracellular concentration of FLC. H99 and *isw1Δ* cells (*n* = 3 each) were incubated in the presence of 40 µg/ml FLC at 30°C for 5 hr. Cells were then washed and weighed, and the intracellular FLC was quantified using high-performance liquid chromatography. A two-tailed unpaired *t*-test was used. Data are expressed as mean ± SD. (**e**) Scheme of the mechanism for 5-FC and 5-fluorouracil (5-FU) resistance. The red arrow indicates the entry of 5-FC via Fcy2. Green arrows indicate the downregulation of gene expression in response to 5-FC in the *isw1Δ* strain. (**f**) Analyses of 5-FC-resistance genes using quantitative reverse transcription PCR(qRT-PCR). Indicated strains (*n* = 3 each) were grown with or without 400 µg/ml 5-FC, then qRT-PCR was performed to determine gene expressions of *FCY1*, *FUR1*, *UGD1*, *UXS1*, and *FCY2*. Two-tailed unpaired *t*-tests were used. Data are expressed as mean ± SD. (**g**) Chromatin immunoprecipitation (ChIP)-PCR analysis of Isw1-FLAG. *ISW1-FLAG* cells (*n* = 3) were incubated without drug treatment at 30°C for 5 hr. ChIP-PCRs were performed in the *ISW1-FLAG*

*Figure 1 continued on next page*

*Figure 1 continued*

strain (*n* = 3 each). Isw1 target genes, *INO1* and *INO80,* were amplified and used as positive controls. *CNAG_06338* were used as a negative control. Potential target genes, *CNAG_03600, UXS1, UGD1, FCY1,* and *FCY2* were tested. Two-tailed unpaired *t*-tests were used. Data are expressed as mean ± SD. (**h**) Truncation analysis of cryptococcal Isw1. Truncated Isw1s were cloned and transformed into the *isw1Δ* strain to generate the DNA-binding domain (DBD) truncation (*isw1Δ/ISW1*$^{DBDΔ}$), SNF truncation (*isw1Δ/ISW1*$^{SNFΔ}$) and helicase C truncation (*isw1Δ/ISW1*$^{Helicase-CΔ}$) strains, which were spotted onto YPD agar plates supplemented with indicated antifungal drugs. (**i**) qRT-PCR analysis in truncated strains supplemented with 5-FC or FLC. Indicated strains were treated with 5-FC or FLC. RNA samples were isolated. qRT-PCRs were performed using oligos from *UGD1, UXS1, FCY2, CNAG_03600,* and *CNAG_06338.*

The online version of this article includes the following source data and figure supplement(s) for figure 1:

**Source data 1.** The source data comprises unprocessed data.

**Figure supplement 1.** Construction of the *ISW1-FLAG* complementation strain.

**Figure supplement 1—source data 1.** The source data comprises unprocessed data.

the FLC-treated *isw1Δ* mutant (***Figure 1c***). Furthermore, we conducted a comparative analysis of the transcriptome data acquired from the wild-type and *isw1Δ* mutant strains in the absence of FLC treatment. The evidence presented indicates that the modulation of drug pumps by Isw1 is subject to the influence of FLC. This phenomenon can be attributed to the identification of modified gene expression in 11 specific genes within the *isw1Δ* mutant strain when exposed to FLC, while no alterations were observed in the absence of FLC. Nevertheless, a total of 10 genes displayed modified expression patterns, regardless of any administered drugs. The data collectively indicate that Isw1 plays a crucial role in regulating the drug efflux mechanism during drug treatment. In addition, in order to investigate the potential modulation of intracellular FLC concentration by Isw1, we employed high-performance liquid chromatography to quantify the levels of intracellular FLC. The results revealed a substantial reduction in FLC levels in the *isw1Δ* mutant compared to the wild-type strain (***Figure 1d***).

To further decipher 5-FC-resistance mechanisms of *isw1Δ*, we examined 5-FC-resistance pathways. As previously elucidated, *C. neoformans* employs two molecular processes for resistance to 5-FC and 5-FU (***Billmyre et al., 2020***; ***Loyse et al., 2013***). In one, the purine-cytosine permease Fcy2 imports the prodrug 5-FC, which is then converted to toxic 5-FU by the cytosine deaminase Fcy1 (***Figure 1e***). On the other, the UDP-glucose dehydrogenase Ugd1 and the UDP-glucuronate decarboxylase Uxs1 participate in UDP-glucose metabolism, providing important functions in detoxifying 5-FU (***Figure 1e***). Therefore, gene expression alterations in *FCY1, FCY2, FUR1, UGD1,* and *UXS1* were assayed using qRT-PCR (***Figure 1f***). The data showed that the expression of *FCY1, FCY2, UGD1,* and *UXS1* was significantly reduced in *isw1Δ* when treated with 5-FC, suggesting a reduction in 5-FC uptake and conversion to toxic 5-FU (***Figure 1f***). These data suggested that Isw1 is a master transcriptional regulator of drug-resistance genes in *C. neoformans.*

Given the notable resemblance between cryptococcal Isw1 and the imitation switch chromatin remodeler found in *S. cerevisiae* (***Chacin et al., 2023***; ***Lin et al., 2019***; ***Litwin et al., 2023***), we sought to determine if the modulation of chromatin activity is the mechanism by which Isw1 regulates drug resistance. Initially, an *ISW1-FLAG* strain was generated by integrating the *ISW1-FLAG* construct into the original *ISW1* allele of the *isw1Δ* strain, for the purpose of conducting chromatin immunoprecipitation tests (***Figure 1—figure supplement 1b–d***). The results revealed that the Isw1-Flag exhibited a significant binding affinity toward the promoter regulatory regions of many genes associated with drug resistance (***Figure 1g***). We then identified five distinct protein domains, including DNA-binding domains (DBD), SNF2, helicase C, HAND, and SLIDE domains in Isw1. Functions of these domains in DNA binding or chromatin remodeling processes were well established in *S. cerevisiae* (***Grüne et al., 2003***; ***Mellor and Morillon, 2004***; ***Pinskaya et al., 2009***; ***Rowbotham et al., 2011***). We thereby generated truncation mutant alleles for these domains, specifically DBD, SNF2, or helicase C. The results showed that Isw1 function is significantly impaired (***Figure 1h***). Consistently, these truncation mutant strains exhibited significant resistance to the drug compounds, indicating the critical role of these domains in mediating Isw1 function (***Figure 1h***). In addition, we found that these truncation strains exhibited deficiencies in activating genes associated with drug resistance, mimicking the regulatory patterns observed in the *isw1Δ* strain (***Figure 1i***). Collectively, these data strongly supported the proposition that the chromatin remodeler Isw1 plays a pivotal role as a primary regulator of antifungal drug resistance in *C. neoformans.* This regulatory function is achieved through the direct interaction of Isw1 with its target genes and subsequently alteration of gene expression.

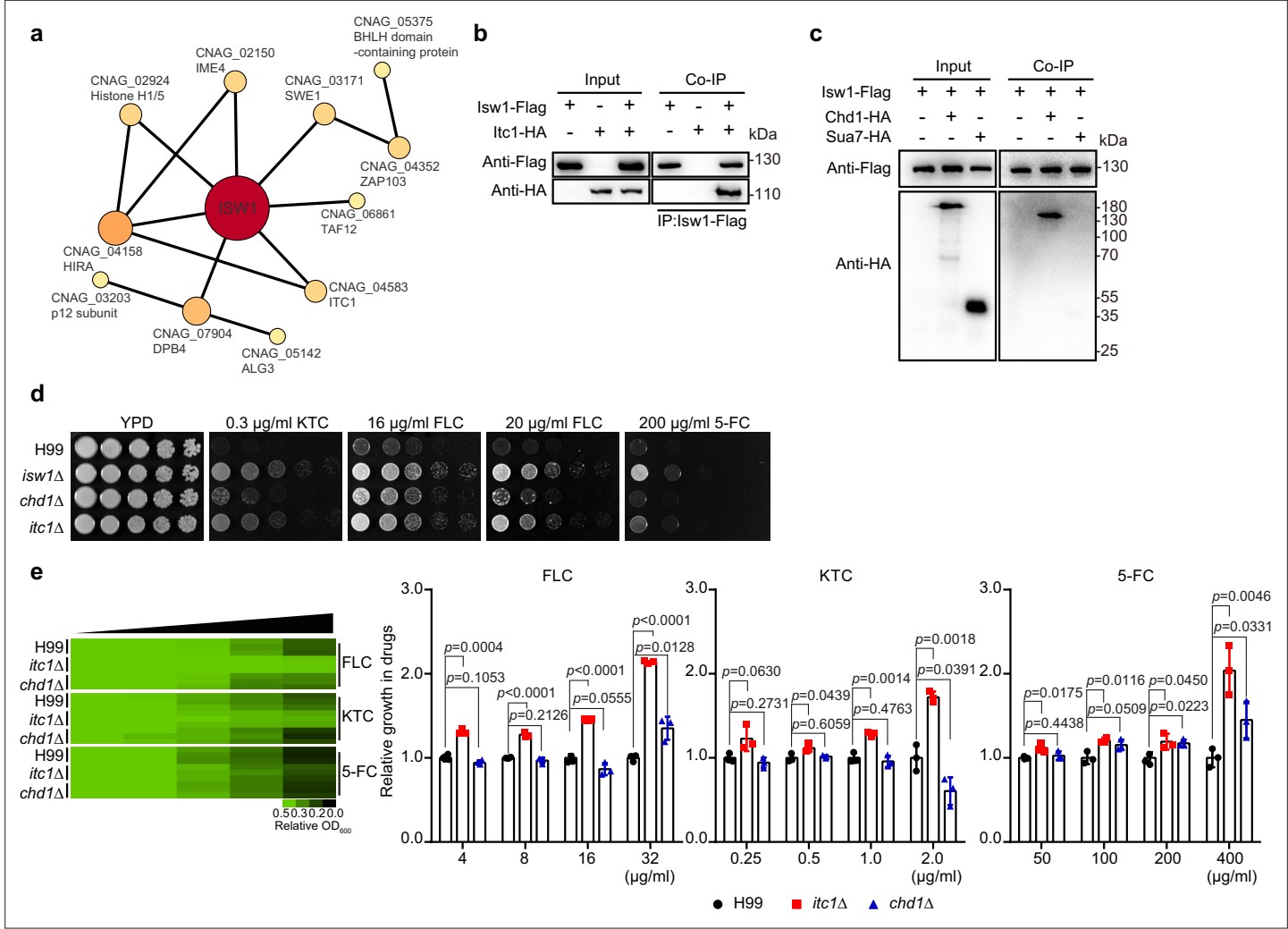

**Figure 2.** Characterization of Isw1-interacting proteins. (**a**) Mass spectrometry analysis of Isw1 regulating network. Protein co-immunoprecipitation (co-IP) assays were carried out, followed by mass spectrometry analysis. Isw1-interacting proteins were identified, and interacting network was generated using StringDB. (**b**) Isw1 and Itc1 co-IP analysis. Proteins were isolated from strains expressing Isw1-FLAG or Itc1-HA or both, and co-IP followed by immunoblotting assays were performed. Antibodies against FLAG and HA epitope tags were used. (**c**) Isw1, Sua1, and Chd1 co-IP analysis. Proteins were isolated from strains expressing Isw1-FLAG, Isw1-FLAG/Chd1-HA, or Isw1-FLAG/Sua7-HA, and co-IP followed by immunoblotting assays were performed. Antibodies against FLAG and HA epitope tags were used. (**d**) Spotting analysis of *chd1Δ* and *itc1Δ* strains. Indicated strains were spotted onto YPD agar plates supplemented with antifungal agents. (**e**) Drug inhibitory tests of *chd1Δ* and *itc1Δ* strains. The H99, *chd1Δ*, and *itc1Δ* strains (*n* = 3 each) were tested to determine inhibitory effects of several antifungal agents. The experiments were performed as described in *Figure 1b*. Two-tailed unpaired *t*-tests were used. Data are expressed as mean ± standard deviation (SD).

The online version of this article includes the following source data for figure 2:

**Source data 1.** The source data comprises unprocessed data.

## Characterization of cryptococcal Isw1-interacting proteins

The ISWI complex in *S. cerevisiae* typically comprises several protein components, such as Isw1, Ioc2, Ioc3, Ioc4, and Itc1. It functions in collaboration with Chd1, which has a substantial overlap in transcriptional target genes with Isw1 (*Mellor and Morillon, 2004*; *Smolle et al., 2012*; *Sugiyama and Nikawa, 2001*; *Vary et al., 2003*; *Yadon et al., 2013*). To further characterize cryptococcal ISWI functional mechanisms and regulatory pathways, we conducted a co-immunoprecipitation (co-IP) assay with Isw1-Flag, followed by mass spectrometry analysis (*Supplementary file 4*). We have found a total of 22 proteins that interacted with Isw1-Flag, and 11 are anticipated to be involved in the Isw1 network (*Figure 2a*), as shown by the STRING database (https://version-11-5.string-db.org). The results showed common Isw1-interacting proteins, including histones (such as histone H1/5,

CNAG_02924) and Itc1. Because the Itc1 subunit has a notable function in the yeast ISWI complex (*Sugiyama and Nikawa, 2001*), we performed the protein–protein interaction between Isw1 and Itc1. As *S. cerevisiae* Isw1 acts in conjugation with Chd1 and Isw2's DNA-binding ability has been implicated to be Sua7 dependent, these interactions were also tested (*Smolle et al., 2012*; *Yadon et al., 2013*). Similar to that of *S. cerevisiae*, Itc1 was also observed to engage in protein–protein interactions in *C. neoformans* (*Figure 2b*). Our mass spectrometry analysis failed to identify a homolog of Chd1 (*Supplementary file 4*). Nevertheless, the Chd1 homolog was found to co-immunoprecipitate with Isw1 in *C. neoformans*, whereas the Sua7 homolog did not exhibit the same manners (*Figure 2c*).

We then investigated the extent to which the regulation of drug resistance by cryptococcal Isw1 is contingent upon the presence of these subunits. We created the mutants of *itc1Δ* and *chd1Δ* (*Figure 2d, e*) and found that *itc1Δ* cells were resistant to antifungal agents, but the *chd1Δ* strain demonstrated only limited drug resistance, which are not comparable to those observed in *isw1Δ* cells (*Figure 2d, e*). Therefore, these findings indicate that the control of Isw1 in drug resistance relies on the Isw1–Itc1 complex. However, given the presence of other Isw1-interacting proteins in our data, it is possible that Isw1 may potentially carry out its function through alternative complex regulatory networks.

## Isw1 is an important *in vivo* drug-resistance regulator

We then sought to ask whether Isw1 plays a role in fungal pathogenicity and drug resistance in the context of cryptococcal infection and antifungal treatment. We gathered the evidence that total Isw1 protein levels are not affected by external stress inducers (*Figure 3—figure supplement 1*). In addition, we observed that the *isw1Δ* strain did not exhibit any growth impairment when subjected to stresses, such as changes in body temperature, osmotic stress, and stresses affecting the cell wall and membrane (*Figure 3a*). Moreover, a minor impairment in melanin production was observed, but no noticeable abnormalities in the development of the capsule were identified (*Figure 3b, c*). Next, cryptococcal intranasal infections were performed in BALB/c mice (*Oliveira et al., 2021*; *Li et al., 2019*). The results revealed no statistically significant changes in the survival rates and colony-forming units (CFUs) (14-day post infection) among the wild-type, *isw1Δ*, and *ISW1* complemented strains (*Figure 3d, e*). Finally, mice were subjected to intranasal infection with either the wild-type or *isw1Δ* cells, followed by intraperitoneal administration of FLC or 5-FC. Pulmonary CFU quantification revealed that the *isw1Δ* strain exhibits enhanced fungal burdens when treated with both drugs (*Figure 3e*). This enhancement is statistically significant, as indicated by the p-values of less than 0.0001 ($p < 0.0001$). These data thus provided substantial evidence supporting the pivotal role of Isw1 as a key regulator of drug resistance, both *in vivo* and *in vitro*.

## Isw1 undergoes protein degradation in the presence of azole compounds

Because Isw1 governs the expression of multiple genes required for azole resistance and the *ISW1* gene expression was not reduced in the presence of antifungal drugs (*Figure 1—figure supplement 1a*), we examined changes in protein stability as a response to antifungal agents. Using cycloheximide to inhibit protein synthesis, the Isw1-Flag fusion protein stability was found to decrease gradually in concentration- and time-dependent manner upon exposure to FLC (*Figure 4a, b*). Specifically, a reduction of 50% was observed 30 min after FLC exposure. Similarly, 5-FC exposure also reduced Isw1-Flag levels (*Figure 4c, d*). Collectively, these results demonstrated that *C. neoformans* actively reduces Isw1 protein levels through protein degradation in the presence of antifungal drugs.

We then hypothesized that Isw1 degradation might be via a ubiquitin–proteasome pathway in response to antifungal drugs. To test this, the Isw1-Flag protein was immunoprecipitated and then analyzed using mass spectrometry to identify putative ubiquitination PTM sites. Consistent with our hypothesis, Isw1 is ubiquitinated (*Figure 4e*) at 15 sites (*Figure 4f*). These results indicated that the ubiquitination machinery of Isw1 is actively initiated during drug exposure, and this, in turn, decreases Isw1 protein levels and hinders Isw1 transcription repression of genes for drug resistance. The finding of Isw1 subject to ubiquitination and acetylation regulation also suggested that there exists an interplay network simultaneously controlling Isw1 stability in response to antifungal drugs. We then set forth to address whether there is a PTM interplay between acetylation and ubiquitination in Isw1.

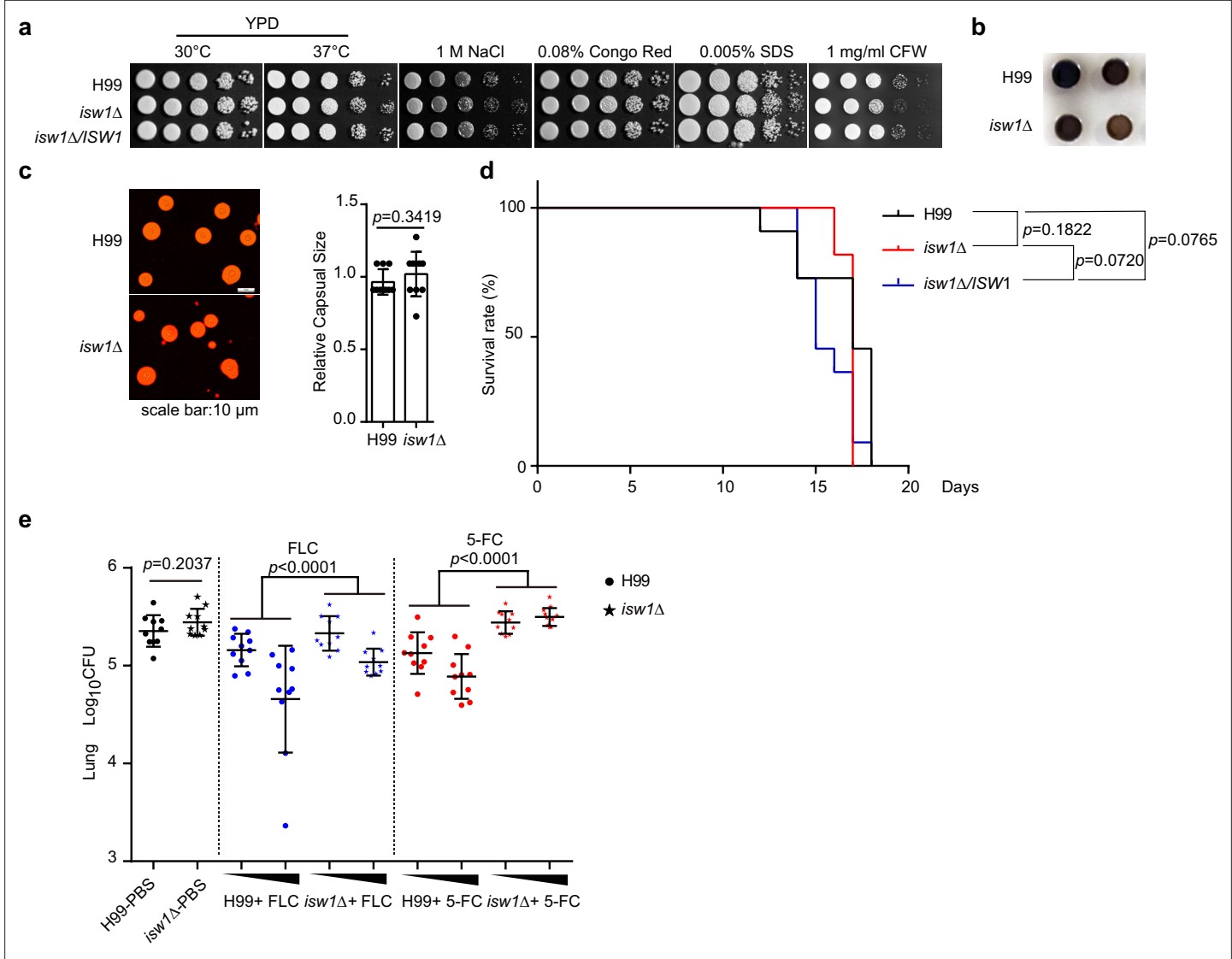

**Figure 3.** Isw1 plays a critical role in drug resistance during pulmonary infection. (**a**) Spotting analysis of *isw1Δ* in stress conditions. H99, *isw*1Δ, and *ISW1* complementation strains were spotted onto YPD agar plates supplemented with indicated chemicals. (**b**) Melanin formation of the *isw1Δ* strain. (**c**) Capsule formation and quantification of the *isw1Δ* strain. H99 and isw1Δ cells were induced for capsule structure. Capsular sizes were quantified. Two-tailed unpaired *t*-tests were used. Data are expressed as mean ± standard deviation (SD). (**d**) Animal survival analysis and the Kaplan–Meier survival curves of wild-type and *isw1Δ*. Significance was determined using a log-rank (Mantel–Cox) test. (**e**) Colony-forming unit (CFU) analysis of the *isw1Δ* strain. Mice (*n* = 10 each) were infected with H99 or *isw1Δ* cells. At 7-day post infection, animals were treated with phosphate-buffered saline (PBS), or 5 or 45 mg/kg of fluconazole (FLC), or 100 or 200 mg/kg 5-fluorocytosine (5-FC). Animals were treated with drugs in a 24-hr interval on a daily basis. At 14-day post infection, lung tissues were removed and homogenized. CFUs were performed. Data are expressed as mean ± standard deviation (SD). Two-tailed unpaired *t*-tests were used for H99-PBS and *isw1Δ*-PBS group. Two-way analyses of variance (ANOVAs) were used for drug-treated groups.

The online version of this article includes the following source data and figure supplement(s) for figure 3:

**Source data 1.** The source data comprises unprocessed data.

**Figure supplement 1.** Isw1 is not responsive to other environmental stresses.

**Figure supplement 1—source data 1.** The source data comprises unprocessed data.

## Acetylation of Isw1$^{K97}$ (Isw1$^{K97ac}$) is essential for protein stability

To dissect the interplay between acetylation and ubiquitination in Isw1, we examined the role of acetylation in modulating Isw1 function by determining acetylation levels responding to antifungal drugs. The presence of antifungal agents strongly repressed acetylation levels, in contrast to deacetylation inhibitors trichostatin A (TSA) and nicotinamide (NAM), which enhanced acetylation levels (*Figure 5a,*

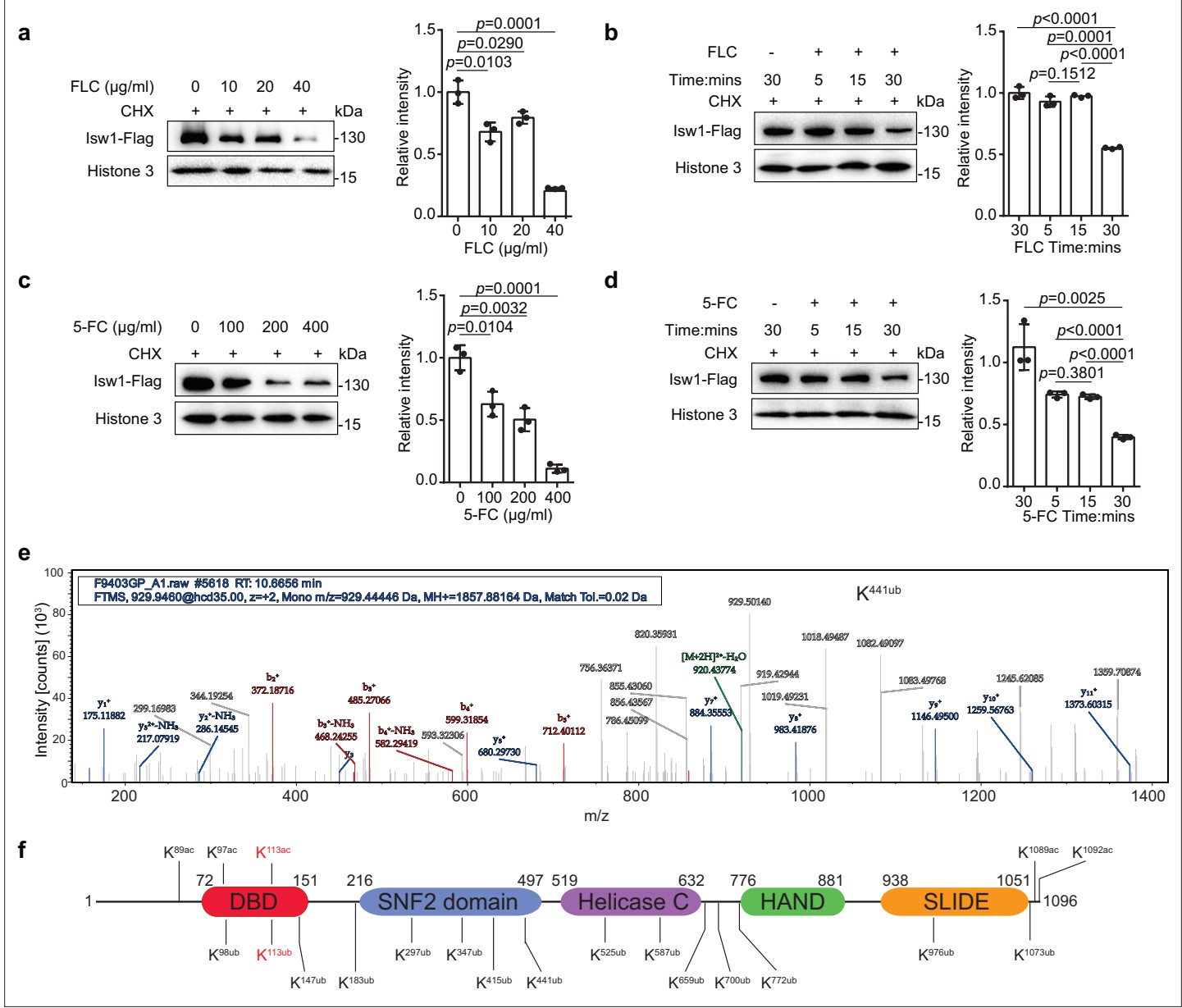

**Figure 4.** Isw1 is an acetylated and ubiquitinated protein. (**a**) Immunoblotting analysis. The *ISW1-FLAG* strain was preincubated with 200 μM cycloheximide (CHX) for 1 hr followed by exposure to various concentrations of fluconazole (FLC) for 0.5 hr. Anti-Flag and anti-histone 3 antibodies were used. Three biological replicates were performed, and results were used for quantification. Two-tailed unpaired *t*-tests were used. Data are expressed as mean ± standard deviation (SD). (**b**) Immunoblotting analysis. The *ISW1-FLAG* strain was preincubated with 200 μM cycloheximide (CHX) for 1 hr followed by exposure to 40 μg/ml FLC for 5, 15, or 30 min. Cells not exposed to FLC but held for 30 min were used as a negative control. Anti-Flag and anti-histone 3 antibodies were used. Three biological replicates were performed, and results were used for quantification. Two-tailed unpaired *t*-tests were used. Data are expressed as mean ± SD. (**c**) Immunoblotting analysis. Testing and data treatment were exactly as described for *Figure 2a* with the exception that 5-fluorocytosine (5-FC) was used as the antifungal agent. (**d**) Immunoblotting analysis. Testing and data treatment were exactly as described in *Figure 2b*, except that 5-FC was used as the antifungal agent. (**e**) Ubiquitin analysis of Isw1 via mass spectrometry. The Isw1-Flag proteins were pulled down and analyzed for ubiquitination. Results for Isw1$^{K441Ub}$ are shown. (**f**) Schematic of Isw1 showing acetylation (*Li et al., 2019*) and ubiquitination sites.

The online version of this article includes the following source data for figure 4:

**Source data 1.** The source data comprises unprocessed data.

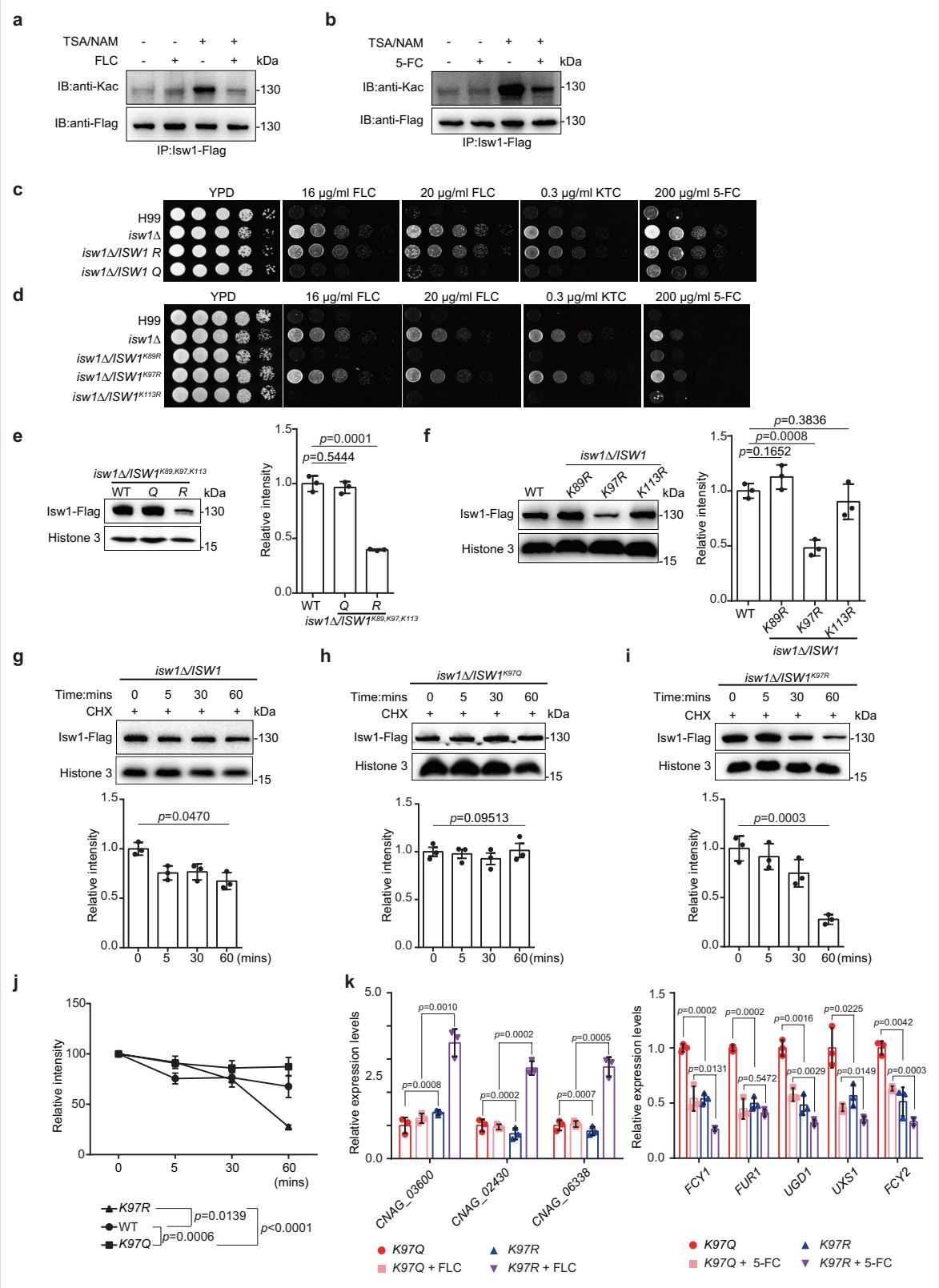

**Figure 5.** The acetylation status of Isw1[K97] (Isw1[K97ac]) is essential in Isw1 protein stability. (**a**) Acetylation analysis of Isw1. Cells were treated with 3 µM trichostatin A (TSA), 20 mM nicotinamide (NAM), and fluconazole (FLC). The Isw1-Flag proteins were pulled down, and immunoblotting assays were performed using anti-Kac and anti-Flag antibodies. (**b**) Acetylation analysis of Isw1. Cells were treated with TSA, NAM, and 5-fluorocytosine (5-FC). The Isw1-Flag proteins were pulled down, and immunoblotting assays were performed using anti-Kac and anti-Flag antibodies. (**c**) Spotting assays of *ISW1*

*Figure 5 continued on next page*

*Figure 5 continued*

mutants. The *ISW1*^K89R, K97R, K113R^ and *ISW1*^K89Q, K97Q, K113Q^ strains were tested for drug resistance. (**d**) Spotting assays of *ISW1* mutants. The *ISW1*^K89R^, *ISW1*^K97R^, and *ISW1*^K113R^ strains were tested for drug resistance. (**e**) Immunoblotting assays of *ISW1* mutants. The wild-type, *ISW1*^K89R, K97R, K113R^, and *ISW1*^K89Q, K97Q, K113Q^ strains were tested for Isw1 levels. Three biological replicates were performed, and results were used for quantification. Two-tailed unpaired *t*-tests were used. Data are expressed as mean ± standard deviation (SD). (**f**) Immunoblotting assays of *ISW1* mutants. The wild-type, *ISW1*^K89R^, *ISW1*^K97R^, and *ISW1*^K113R^ strains were tested for Isw1 levels. Three biological replicates were performed, and results were used for quantification. Two-tailed unpaired *t*-tests were used. Data are expressed as mean ± SD. (**g**) Immunoblotting assay of Isw1. The wild-type strain was preincubated with 200 µM cycloheximide for 1 hr. Proteins were isolated at indicated time points. Three biological replicates of immunoblotting were performed, and results were used for quantification. One-way analysis of variance (ANOVA) was used. (**h**) Immunoblotting assay of Isw1^K97Q^. The analysis was performed as described in *Figure 3g*. (**i**) Immunoblotting assay of Isw1^K97R^. The analysis was performed as described in *Figure 3g*. (**j**) Comparisons of assay results to determine Isw1 stability. The relative intensities from the results shown in *Figure 3g–i* were plotted. Two-way ANOVA was used. Data are expressed as mean ± SD. (**k**) Analyses of drug-resistance genes using qRT-PCR. Samples of RNA (*n* = 3) were isolated from *ISW1*^K97Q^ and *ISW1*^K97R^ treated with FLC or 5-FC. Representative drug-resistance genes were quantified using qRT-PCR. Two-tailed unpaired *t*-tests were used. Data are expressed as mean ± SD.

The online version of this article includes the following source data and figure supplement(s) for figure 5:

**Source data 1.** The source data comprises unprocessed data.

**Figure supplement 1.** Screening important acetylation sites of Isw1.

**Figure supplement 1—source data 1.** The source data comprises unprocessed data.

*b*). These data suggested a positively regulated deacetylation process in Isw1 in response to antifungal drugs. To decipher this regulation mechanism more, three acetylation sites, K89, K97, and K113, located to the DBD were mutated to arginine (R) to mimic a deacetylated status or to glutamine (Q) to mimic acetylated Isw1 (*Figure 4f*). Gene copy numbers and transcription levels were confirmed to be equivalent to those of the wild-type strain (*Figure 5—figure supplement 1a, b*). Triple-, double-, and single-mutated strains were generated, and their drug-resistance phenotypes were compared. Of the triple-mutated strains, cells with three R mutations demonstrated drug-resistant growth phenotypes that were similar to those of the *isw1Δ* strain, whereas those with three Q mutations showed the wild-type growth in the presence of antifungal drugs (*Figure 5c*). Double-mutated strains, *ISW1*^K89R, K97R^ and *ISW1*^K97R, K113R^, showed resistance to antifungal drugs, wherein the *ISW1*^K97R, K113R^ strain showed less resistance than *ISW1*^K89R, K97R^ and the *isw1Δ* strain (*Figure 5—figure supplement 1c*). These data suggested that the acetylation status of Isw1^K97^ is important in conferring drug resistance. Of the strains that have single-R mutation, the *ISW1*^K97R^ strain showed robust resistance to antifungal drugs, mimicking the *isw1Δ* strain (*Figure 5d*). Interestingly, the *ISW1*^K97Q^ strain showed no drug resistance (*Figure 5—figure supplement 1d*). Collectively, these data strongly demonstrated that the acetylation status of Isw1^K97^ plays a critical role in regulating Isw1 protein stability and function in response to antifungal drugs.

To further investigate how Isw1 degradation correlates with drug resistance of *C. neoformans*, we tested how Isw1^K97^ acetylation affects its degradation using the immunoblotting method, and the results showed that triple-R mutation resulted in a significant reduction in levels of Isw1-Flag. Meanwhile, triple-Q mutation resulted in Isw1-Flag levels comparable to wild-type strains (*Figure 5e*). Similarly, the single site mutant, *ISW1*^K97R^, showed a lower level of Isw1-Flag (*Figure 5f*). In contrast, no changes were observed for Isw1^K97Q^ (*Figure 5—figure supplement 1e*). Moreover, protein levels of wild-type Isw1 and mutated Isw1^K97R^ gradually diminished over time. While those of Isw1^K97Q^ remained constant (*Figure 5g–i*), a faster degradation was observed for Isw1^K97R^ (*Figure 5j*). Therefore, Isw1^K97^ is an essential regulation site responsible for Isw1 stability; that is, acetylation at K97 blocked the degradation of Isw1, and deacetylation at K97 facilitated Isw1 degradation. Finally, analysis of Isw1 target gene expression in Isw1^K97^ mutation strains demonstrated significantly the increased expression of transporter genes and the decreased expression of 5-FC-resistance genes (*Figure 5k*). These findings were consistent with the results of transcriptome and qRT-PCR analyses in the *isw1Δ* strain (*Figure 1c, f*).

## The interplay between acetylation and ubiquitination governs Isw1 degradation

As proteins undergo degradation via autophagy and proteasomal pathways, we employed autophagy inhibitor rapamycin and proteasome inhibitor MG132 to investigate Isw1 degradation. Immunoblotting results showed that rapamycin and MG132 promote Isw1-Flag levels, with rapamycin's effect

less prominent than MG132 (*Figure 6a*). These findings suggested that both degradation pathways are utilized in Isw1-Flag degradation, but the ubiquitin-mediated proteasomal process has a more predominant role. Additionally, we tested whether antifungal agents could induce protein degradation when the proteasome pathway is blocked. Cells treated with MG132 and FLC or 5-FC yielded slightly different results. Isw1-Flag protein levels were unaffected in cells treated with FLC, but the levels were reduced with 5-FC (*Figure 6b, c*).

Given that K97 deacetylation could trigger hyper-ubiquitination of Isw1, we analyzed Isw1 ubiquitination sites and their regulation mechanisms by K97 acetylation levels and found that MG132 treatment results in a more robust enhanced in Isw1$^{K97R}$-Flag levels (*Figure 6d*). We further performed ubiquitination site mutations in the genetic background of *ISW1$^{K97R}$*, and sites neighboring K97 were selected for mutagenesis. We found that, of the seven ubiquitination sites (*Figure 6—figure supplement 1a, b*), five (K98, K147, K183, K347, and K415) failed to affect drug-resistant growth phenotypes of the *ISW1$^{K97R}$* mutant (*Figure 6—figure supplement 1c*). Only *ISW1$^{K113R}$* and *ISW1$^{K441R}$* mutations exhibited reduced drug-resistant growth of the *ISW1$^{K97R}$* stain (*Figure 6e*), indicating that they affect drug resistance by modulating Isw1 protein stability in the absence of acetylation at the K97. The immunoblotting analysis further showed that, while all ubiquitination mutants exhibited moderately enhanced Isw1-Flag levels (*Figure 6f, g*), robust elevations were detected in Isw1$^{K97R, K113R}$ (5.6-fold) and Isw1$^{K97R, K441R}$ (14.5-fold) (*Figure 6f, g*). Interestingly, the K113 site may undergo acetylation or ubiquitination modifications, whereas the K441 site undergoes only ubiquitination. Collectively, these results showed that Isw1$^{K113}$ and Isw1$^{K441}$ provide a predominant role in regulating the ubiquitin-proteasome process of Isw1 and that acetylation at Isw1$^{K97}$ has a broad role in controlling the ubiquitination process at those sites.

## The acetylation status of Isw1$^{K97}$ modulates the binding of an E3 ligase to Isw1

To analyze the molecular mechanism by which Isw1$^{K97}$ regulates the ubiquitin–proteasome process, we generated nine knockout mutants of E3 ligase encoding genes in the genetic background of *ISW1$^{K97R}$* and identified that Cdc4 is an E3 ligase for Isw1 (*Figure 7—figure supplement 1a* and *Figure 7a*). We found that the *ISW1$^{K97R}$/cdc4Δ* strain becomes sensitive to antifungal agents (*Figure 7a*). In addition, disrupting *CDC4* in the H99 strain does not affect the growth of cells in response to antifungal drugs (*Figure 7—figure supplement 1b*). Immunoblotting showed a strong elevation in Isw1$^{K97R}$-Flag levels in the *ISW1$^{K97R}$/cdc4Δ* strain (*Figure 7b*), in contrast to the *ISW1$^{K97R}$/fwd1Δ* strain (*Figure 7c*), suggesting a potential interaction between Cdc4 and Isw1. We then performed a co-IP assay and found that Cdc4-HA co-precipitates with Isw1-Flag (*Figure 7d*).

We also carried out co-IP to examine interactions between Cdc4-HA and Isw1$^{K97R}$ and Cdc4-HA and Isw1$^{K97Q}$, and the results showed an interaction between Cdc4 and Isw1$^{K97R}$ but a reduction in the strength of the interaction between Cdc4 and Isw1$^{K97Q}$, indicating the acetylation of Isw1$^{K97}$ hinders its binding of the Cdc4 E3 ligase (*Figure 7e*). These data provided convincing evidence that K97 acetylation is a key player in modulating ubiquitin–proteasome degradation of Isw1.

## The Isw1–proteasome regulation axis promotes drug resistance in clinical isolates

We have presented empirical evidence that highlights the significant involvement of Isw1 and the acetylation–Isw1–ubiquitination regulatory pathway in the modulation of drug resistance in *C. neoformans*. In order to assess the potential wide-ranging applicability of this regulatory function, a random selection of 18 clinical isolates of *C. neoformans* was subjected to testing (*Supplementary file 1*). Among them, a total of 12 isolates (CDLC4, CDLC13, CDLC15, CDLC25, CDLC60, CDLC61, CDLC62, CDLC98, CDLC125, CDLC135, CDLC141, and CDLC150) exhibited significant resistance to at least one antifungal drug. It is noteworthy that the application of PCR in conjunction with sequencing analysis has yielded results indicating the absence of mutations in the drug-resistance genes of these clinical isolates (*Supplementary file 5*). Conversely, six isolates (CDLC120, CDLC6, CDLC27, CDLC37, CDLC43, and CDLC100) demonstrated no resistance to antifungal agents, and their growth patterns resembled that of the H99 strain in the presence of these drugs (*Figure 8—figure supplement 1a*). Subsequently, a 3' integrative FLAG-tag construct was introduced into multiple clinical strains, enabling the transcription of Isw1-Flag under the control of the *ISW1* endogenous promoter (*Sun*

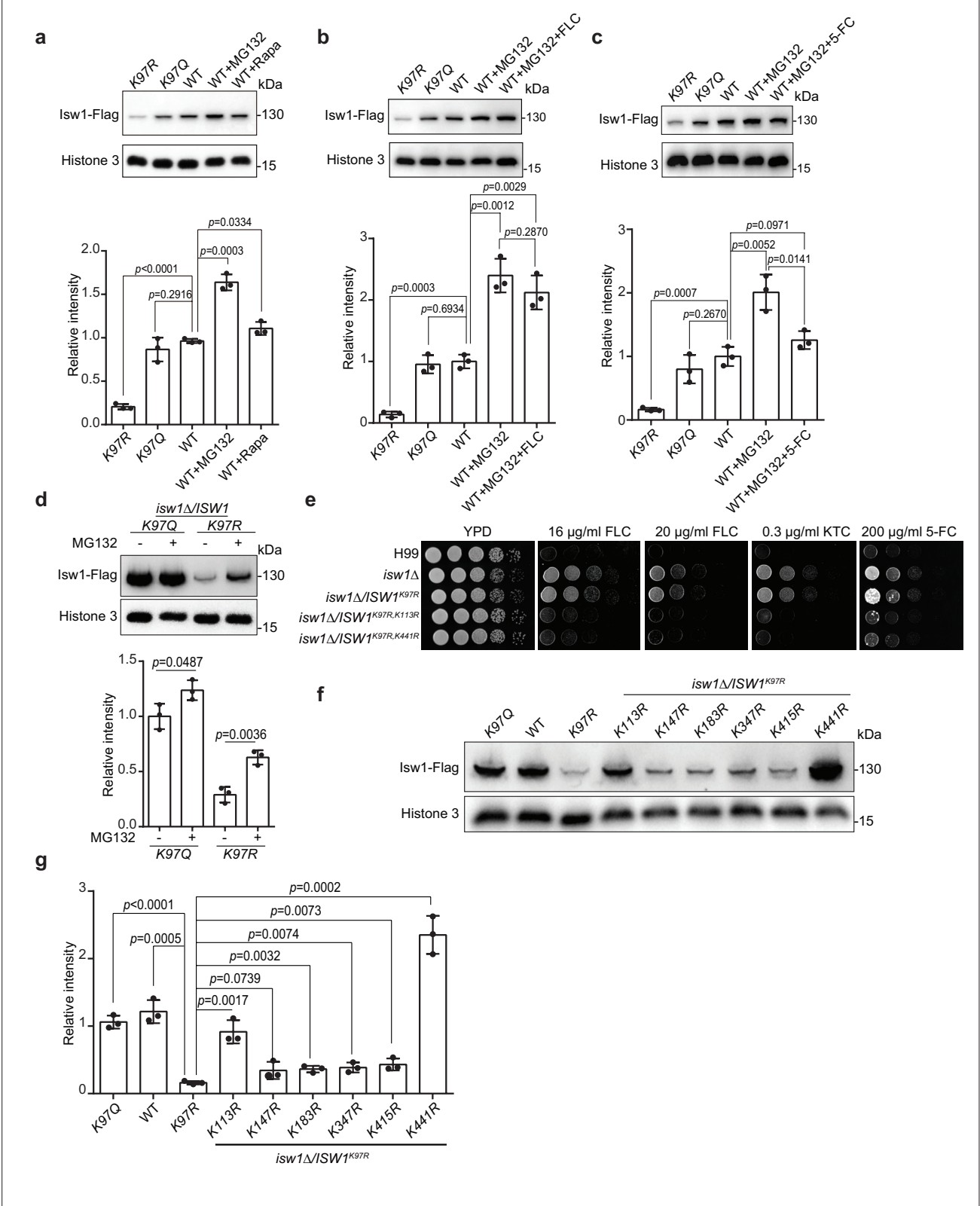

**Figure 6.** Isw1^(K97ac) is critical for Isw1 ubiquitin–proteasome degradation. (**a**) Immunoblotting assays of Isw1-Flag. Proteins Isw1^(K97Q), Isw1^(K97R), and Isw1 were tested, where the wild-type stain was incubated with 200 µM MG132 and 5 nM rapamycin for 10 hr. Three biological replicates of the assays were performed, and results were used for quantification. Two-tailed unpaired *t*-tests were used. Data are expressed as mean ± standard deviation (SD). (**b**) Immunoblotting assays of Isw1-Flag. Testing and data treatment were exactly as described in *Figure 4a*, except that the wild-type sample was

*Figure 6 continued on next page*

*Figure 6 continued*

treated with 200 µM of MG132 and 40 µg/ml fluconazole (FLC). (**c**) Immunoblotting assays of Isw1-Flag. Testing and data treatment were exactly as described in *Figure 4a*, except that the wild-type sample was treated with 200 µM MG132 and 400 µg/ml 5-fluorocytosine (5-FC). (**d**) Immunoblotting assays of Isw1$^{K97Q}$ and Isw1$^{K97R}$. Proteins were either treated with MG132 or not before testing. Three biological replicates of the assays were performed, and results were used for quantification. Two-tailed unpaired *t*-tests were used. Data are expressed as mean ± SD. (**e**) Spotting assays of *ISW1* acetylation and ubiquitination mutants. Indicated strains were spotted onto YPD agar either supplemented with an antifungal agent or left blank. (**f**) Immunoblotting assays of *ISW1* acetylation and ubiquitination mutants. Protein samples were isolated from the indicated *ISW1* mutants. Immunoblotting assays were performed. (**g**) Quantification of immunoblotting results. The immunoblotting assays described for *Figure 3f* were performed using three independent samples, and the results were used for quantification. Two-tailed unpaired *t*-tests were used. Data are expressed as mean ± SD.

The online version of this article includes the following source data and figure supplement(s) for figure 6:

**Source data 1.** The source data comprises unprocessed data.

**Figure supplement 1.** Screening important ubiquitination sites of Isw1.

**Figure supplement 1—source data 1.** The source data comprises unprocessed data.

*et al., 2014*). The protein expression results exhibited three distinct classes: (1) Strains that displayed phenotypes of multidrug resistance exhibited notably diminished levels of Isw1-Flag. These strains include CDLC15, CDLC25, CDLC61, CDLC62, and CDLC98. (2) Conversely, strains that exhibited sensitivity to drug treatment displayed robust protein levels of Isw1-Flag. Notable examples of such strains are CDCL120, CDCLC6, CDCL37, CDCLC43, and CDLC100. (3) Drug-resistant strains demonstrated Isw1-Flag levels comparable to those of the H99 strain. This similarity was observed in strains such as CDCL141 and CDLC4 (*Figure 8a* and *Figure 8—figure supplement 1b*). Significantly, the changes in Isw1-Flag levels observed were not attributed to transcriptional modifications (*Figure 8—figure supplement 1c*).

We then determined whether the Isw1 of clinical isolates exhibit a comparable PTM regulation pattern as observed in the H99 strain. In a manner akin to PTMs observed in Isw1 from the H99 strain, our analysis of samples derived from three distinct clinical isolates revealed heightened levels of Isw1 upon exposure to MG132 (*Figure 8b, c*). Furthermore, the clinical isolates that were subjected to TSA and NAM treatment exhibited enhanced levels of Isw1 acetylation, as demonstrated in (*Figure 8d*). The clinical isolates were subsequently analyzed to ascertain if drug resistance arises due to alterations in Isw1 protein levels subsequent to transformation with an integrative plasmid that overexpresses the non-degradable variant of Isw1, known as Isw1$^{K97Q}$. Initially, the H99 strain was subjected to testing, which revealed a decrease in cell growth when the *ISW1$^{K97Q}$* gene was overexpressed in the presence of antifungal drugs (*Figure 8e*). Subsequently, four strains, namely CDLC120, CDLC141, CDLC135, and CDLC61, were subjected to transformation and subsequent testing. The results indicated that all four clinical strains overexpressing *ISW1$^{K97Q}$* displayed a susceptible cell growth response to antifungal drugs (*Figure 8f–i*). Conversely, the CDLC135 *ISW1$^{K97Q}$* strain exhibited a sensitive growth pattern toward azoles but demonstrated a resistant growth phenotype when exposed to 5-FC (*Figure 8g*). While the Isw1 level remains constant in CDLC141, an elevation in Isw1 level was found to suppress the growth of drug resistance in this strain (*Figure 8h*). Hence, the acetylation–Isw1–ubiquitination regulatory axis represents a naturally occurring mechanism employed to modulate multidrug resistance in clinical strains of *C. neoformans*.

## Discussion

Fungi have developed sophisticated machinery to combat various stress inducers, and the rapid emergence of resistance to antifungal agents is one of the major factors in the failure of clinical therapies for fungal infections (*Denning and Bromley, 2015*). A typical tactic used by fungi to overcome antifungal toxicity is to utilize polymorphisms or mutations in drug targets or their regulatory components. Clinical polymorphisms were widely shown in drug targets, such as ergosterol biosynthesis and its transcription regulatory process (*Denning and Bromley, 2015*; *Hu et al., 2017*). However, unlike immediate intracellular responses, the accumulation of mutations or polymorphisms in drug resistance is a somewhat delayed process that frequently develops over a series of cell divisions. Acetylation and ubiquitination are critical modulators of protein activities or stabilities in fungi that enable rapid intracellular adaptations to environmental or chemical stressors (*Li et al., 2019*; *Wu et al., 2021*), but the

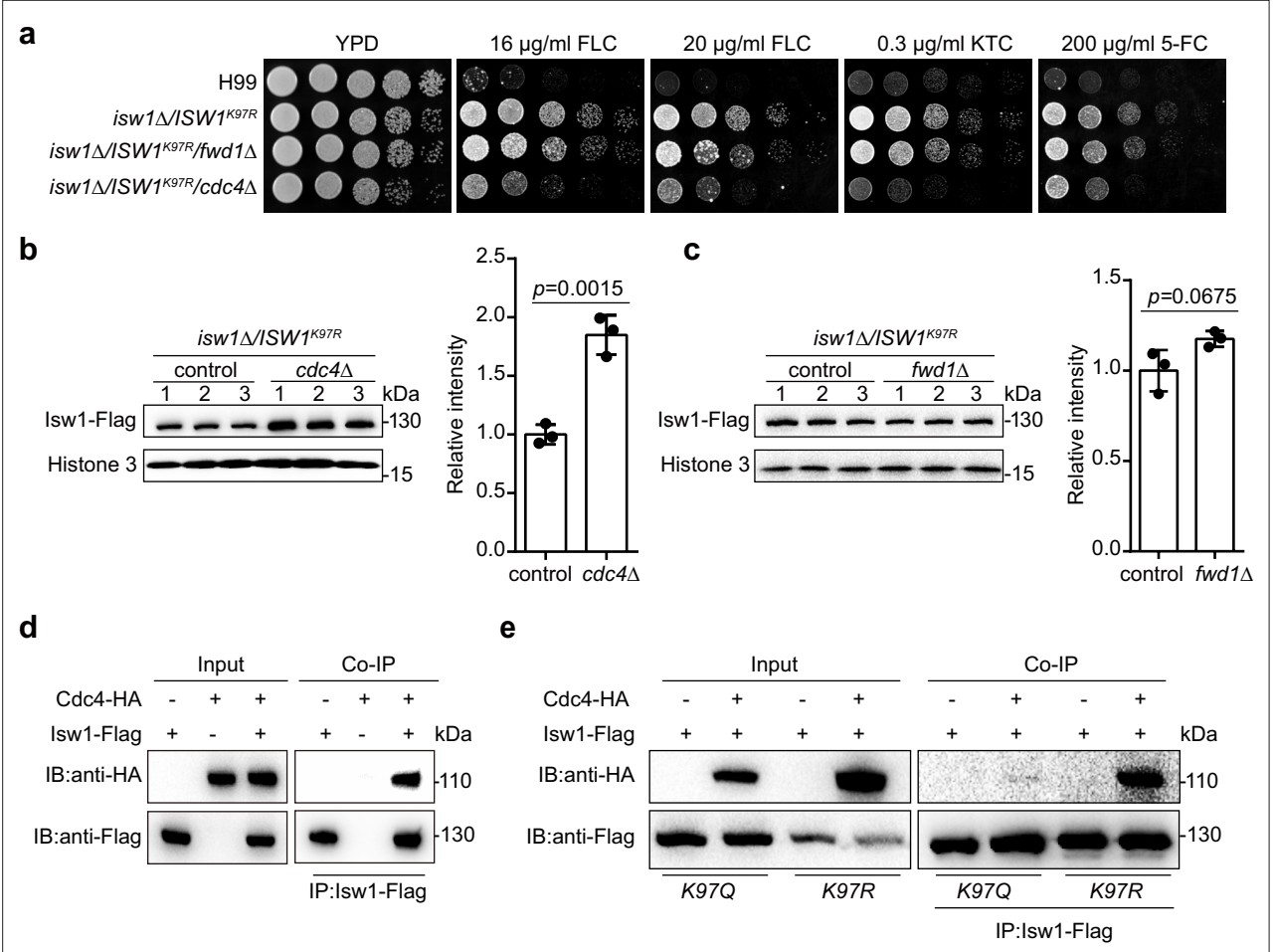

**Figure 7.** Isw[K97ac] blocks the binding of Isw1 to the E3 ligase Cdc4. (**a**) Spotting assays of E3 ligase mutants. Indicated strains were spotted onto YPD agar either supplemented with an antifungal agent or left blank. (**b**) Immunoblotting assays of *cdc4Δ*. Protein samples were isolated from *isw1Δ/ISW1[K97R]/cdc4Δ* and its relevant control strains. Three independent samples were tested and quantified. Two-tailed unpaired *t*-tests were used. Data are expressed as mean ± standard deviation (SD). (**c**) Immunoblotting assays of *fwd1Δ*. Protein samples were isolated from *isw1Δ/ISW1[K97R]/fwd1Δ* and its relevant control strains. Three independent samples were tested and quantified. Two-tailed unpaired *t*-tests were used. Data are expressed as mean ± SD. (**d**) Protein co-immunoprecipitation (co-IP) of Cdc4 and Isw1. Protein samples were isolated from the strain expressing Cdc4-HA and Isw1-Flag, and co-IP was performed. (**e**) Protein co-IP of Cdc4 and Isw1 K97 mutant proteins. Protein samples were isolated from the strain co-expressing Cdc4-HA and Isw1[K97R]-Flag and the strain co-expressing Cdc4-HA and Isw1[K97Q]-Flag. Co-IP was performed for each.

The online version of this article includes the following source data and figure supplement(s) for figure 7:

**Source data 1.** The source data comprises unprocessed data.

**Figure supplement 1.** Identification of the E3 ligase for Isw1.

underlying mechanisms are unclear. Recently, a study showed that the deactivation of a heat-shock Hsp90 client protein and its stability due to changes in protein acetylation impacts drug resistance in *C. albicans* (*Robbins et al., 2012*). Additional studies have demonstrated that the catalytic subunit of the histone acetyltransferase, notably Gcn5, controls biofilm formation, morphology, and susceptibility to antifungal drugs in several fungi (*O'Meara et al., 2010*; *Rashid et al., 2022*; *Yu et al., 2022*). Despite this, knowledge of the molecular machinery of PTMs in modulating drug resistance remains not clear.

We demonstrated that the chromatin remodeler Isw1 is a master regulator of drug resistance in *C. neoformans,* and the acetylation–Isw1–ubiquitination axis is crucial in modulating the expression of multiple drug-resistance genes. In *S. cerevisiae,* Isw1 is a key component of the ISWI complex capable of forming complexes with Ioc2, Ioc3, Ioc4, and Itc1 to modulate transcription initiation and elongation (*Mellor and Morillon, 2004*; *Smolle et al., 2012*; *Sugiyama and Nikawa, 2001*; *Vary et al.,*

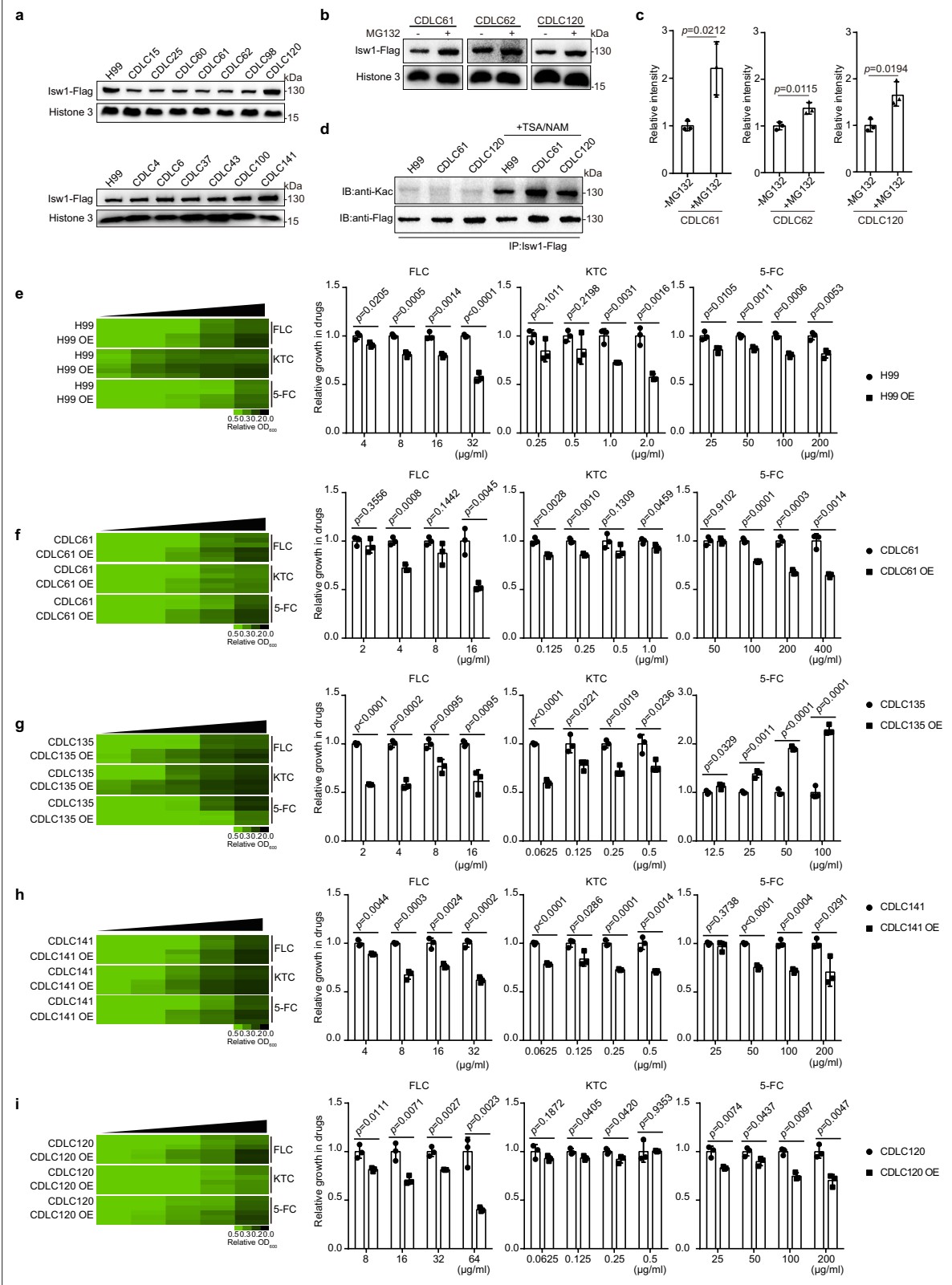

**Figure 8.** Clinical *C. neoformans* isolates show Isw1-mediated drug-resistance phenotypes. (**a**) Immunoblotting assays of Isw1-Flag from clinical isolates. Protein samples were isolated from H99 and clinical isolates expressing Isw1-Flag. Immunoblotting analyses were performed. (**b**) Immunoblotting analyses of Isw1 levels under MG132 treatment. Cells were treated with MG132, then immunoblotting assays were performed on Isw1-Flag using anti-Flag antibodies. (**c**) Three independent repetitions from *Figure 8b* were performed, and results were used for quantification. Two-tailed unpaired *t*-tests

*Figure 8 continued on next page*

*Figure 8 continued*

were used. Data are expressed as mean ± standard deviation (SD). (**d**) Immunoblotting analyses of Isw1 acetylation in clinical strains. Cells were treated with trichostatin A and nicotinamide, then Isw1-Flag was immunoprecipitated. Acetylation levels were examined using anti-Kac antibodies. (**e–i**) Drug inhibitory analyses of Isw1-overexpressing clinical strains. Clinical strains harboring integrative overexpressing plasmid of ISW1K97Q were tested for drug resistance, and minimum inhibitory concentrations (MICs) were determined. OE indicates strains with overexpressed ISW1K97Q. The experiments were performed as described in *Figure 1b*. Quantitative data were depicted in color (see color bar) and bar plots. Two-tailed unpaired *t*-tests were used. Data are expressed as mean ± SD.

The online version of this article includes the following source data and figure supplement(s) for figure 8:

**Source data 1.** The source data comprises unprocessed data.

**Figure supplement 1.** Drug-resistance analysis of clinical isolates.

**Figure supplement 1—source data 1.** The source data comprises unprocessed data.

*2003*; *Yadon et al., 2013*), and *S. cerevisiae* Isw1 and Chd1 exhibit functional overlap in transcription within the Set2 pathway (*Smolle et al., 2012*). Additionally, we ascertain the involvement of subunits of the ISWI complex-related protein in the regulation of drug resistance. Although no *IOC* genes of the yeast Isw1 regulatory machinery are found in the *C. neoformans* genome (https://fungidb.org/fungidb/app), we demonstrated that Chd1 and Itc1 have the ability to form a protein complex with Isw1. Interestingly, the disruption of *ITC1*, but not *CHD1*, resulted in multidrug resistance in *C. neoformans*. This finding provides additional evidence that the regulatory mechanisms of the cryptococcal ISW1 complex in drug resistance are mediated by the Isw1–Itc1 regulatory axis, rather than the Isw1–Chd1 pathway.

In *C. neoformans*, transcriptome analysis revealed 1275 genes, approximately 18.3% of the genome, that were significantly differentially expressed in the *isw1Δ* mutant when treated with FLC. The changes in gene expression had a significant impact on the drug efflux system through the activation of 12 drug pump genes and the repression of 9 pump genes, which included ABC and MFS transporter genes. The results presented suggests that the strong drug-resistance phenotype observed in the *isw1Δ* strain may be attributed to the concurrent modulation of gene expression of multiple drug pumps, which subsequently facilitate the active removal of intracellular drug molecules. The expressions of genes required for resisting 5-FC and 5-FU were reduced when cells were treated with 5-FC. While Isw1 represses genes responsible for resistance to FLC, it also positively modulates the expression of genes required for 5-FC resistance, implying that chromatin remodeling of Isw1 is necessary as the cell responds to disparate chemical stresses and that Isw1 engages distinct remodeling venues to overcome drug toxicity. The overall abundance of Isw1 remains constant in response to external stimuli, and alterations in Isw1 protein levels may change when protein synthesis is inhibited, due to the interplay between two PTMs. Nevertheless, the investigations on the growth of cells demonstrated that the *isw1Δ* strain exhibited the ability to endure stresses, so indicating that Isw1 does not regulate fungal virulence factors and pathogenicity. The fungal burden analysis provided further evidence by demonstrating comparable amounts of fungal colonization in both the wild-type and *isw1Δ* strains. However, the *isw1Δ* strain displayed notable resistance to antifungal treatments both *in vitro* condition and in animal models. The findings provide strong evidence for the pivotal function of Isw1 in the development of drug resistance in *C. neoformans* during infection, and revealed a previously unidentified controller of drug resistance in fungi.

We also showed that the Isw1 protein and its acetylation level act reciprocally to govern drug resistance in *C. neoformans* (*Figure 9*). This was confirmed by uncovering the interplay mechanism between acetylation and ubiquitination. The total acetylation levels of Isw1 were reduced when cells were treated with FLC or 5-FC, leading to the activation of Isw1 ubiquitination machinery. We identified that the K97 acetylation site functions as the essential regulating component of this interplay. We also found that K97 acetylation acts as a switch for ubiquitin conjugation proceeding proteasome-mediated degradation. When deacetylated, K97 triggers the activation of Isw1 degradation via the ubiquitin–proteasome process (acetylated K97 blocks the physical interaction with Cdc4). Compared to the *ISW1^{K97R}* strain, *ISW1^{K97R}/cdc4Δ* was sensitive to antifungal agents; however, it showed moderately resistant growth in comparison to the wild-type strain. These data suggested that Isw1 could also be modulated by other proteins, such as another uncharacterized E3 ligase. Such hypothesis is supported by evidence from the comparison of Isw1 ubiquitination mutants with those of the

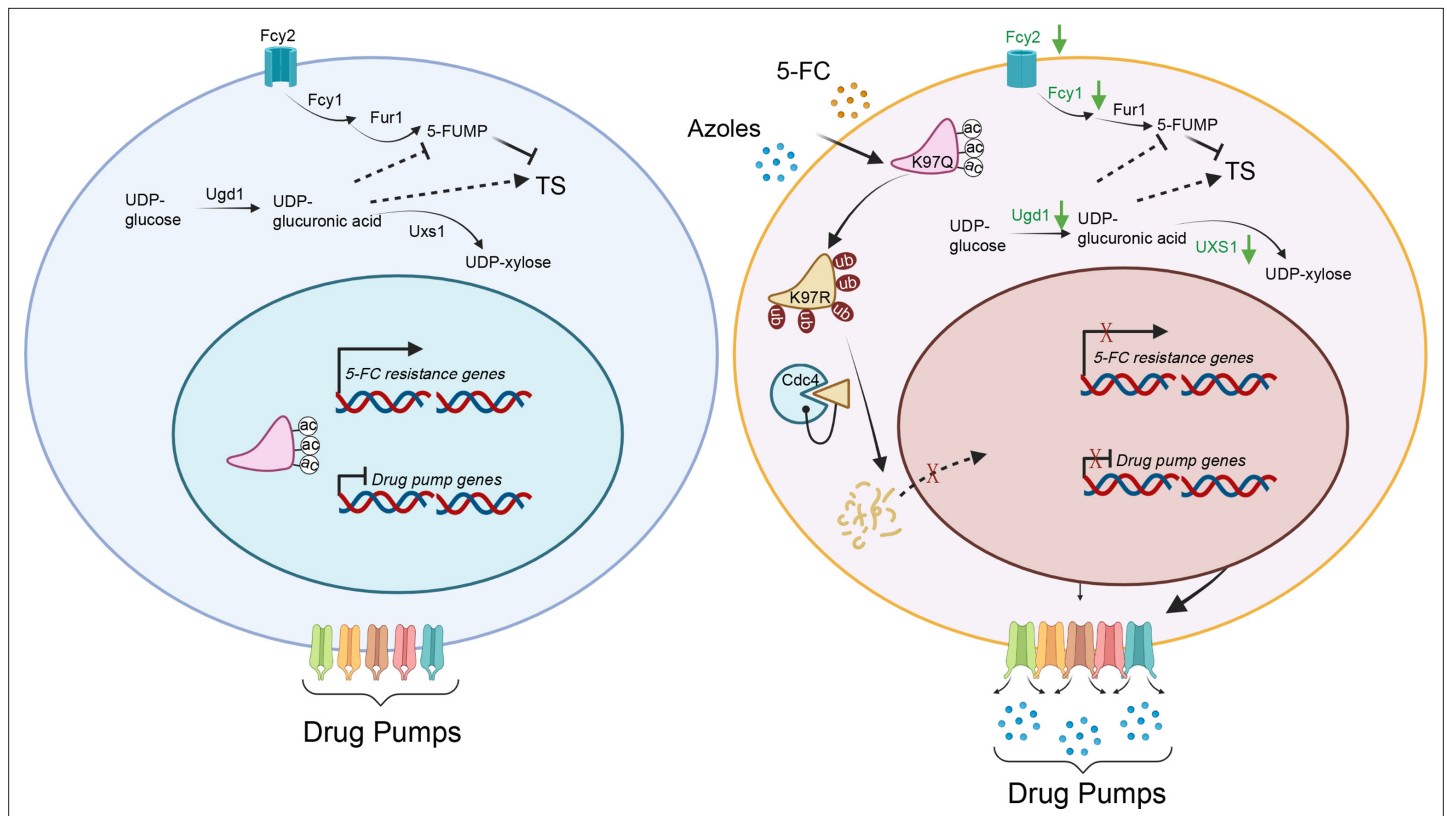

**Figure 9.** A model of the mechanism of Isw1 posttranslational modification (PTM) interaction in *C. neoformans* drug resistance. In a drug-free environment, acetylated Isw1 regulates 5-fluorocytosine (5-FC)-resistance gene expression and represses drug pump gene expression. Azoles and 5-FC trigger the deacetylation process at the K97 residue of Isw1, initiating the ubiquitin-mediated proteasomal degradation of Isw1 through the E3 ligase Cdc4. The decrease in Isw1 protein level results in the stimulation of drug pump gene expression and the inhibition of 5-FC-resistance genes.

$ISW1^{K97R}/cdc4\Delta$ strain, and from that, the $Isw1^{K441R}$ mutant had a 14.5-fold increase of Isw1 and a 2-fold increase in the $ISW1^{K97R}/cdc4\Delta$ strain.

The identified ubiquitination sites were classified into three groups based on the domination of Isw1 stability regulation: predominant, moderate, and minor. While $Isw1^{K441}$ was a predominant regulating site of ubiquitination that extensively modulates Isw1 protein degradation and drug resistance, $Isw1^{K147}$, $Isw1^{K183}$, $Isw1^{K347}$, and $Isw1^{K415}$ played a minor role in Isw1 degradation and no roles in drug resistance. Interestingly, $Isw1^{K113}$ played a moderate role in ubiquitination-mediated Isw1 degradation and was identified to have both an acetylation site and a ubiquitination site. A single mutation at $Isw1^{K113}$ had no effect on Isw1 protein levels since K97 acetylation that prevented Cdc4 binding remains intact. In the strain background of degradable Isw1 mutant ($Isw1^{K97R}$), the K113R mutation was capable of protecting from degradation, and its protein levels had an increase of 5.6-fold, allowing the $ISW1^{K97R, K113R}$ strain to be drug resistant. These data suggested that the K113 site contributes to Isw1 protein stability when the degradation of Isw1 process is initiated. Although it is challenging to dissect the function of acetylation and ubiquitination at $Isw1^{K113}$, drug treatment represents a deacetylation process for Isw1, and deacetylated $Isw1^{K113}$ is most likely ubiquitinated.

The examination of clinical isolates has uncovered a widespread regulatory phenomenon involving Isw1 in the context of drug resistance. In general, clinical isolates exhibiting higher levels of antifungal resistance were shown to have correspondingly reduced expression of Isw1. While certain resistant isolates, including CDCL141, exhibited unaltered endogenous levels of Isw1, the strain's drug-resistance phenotype is contingent upon the levels of Isw1, as evidenced by the heightened sensitivity to antifungal agents upon overexpression of Isw1. Furthermore, acetylation and ubiquitination were detected in the clinical isolates examined, indicating that Isw1 undergoes acetylation and is targeted by the ubiquitin–proteasome system. Significantly, the overexpression of Isw1 was found to mitigate resistance to drugs in clinical isolates. The strain CDLC135 exhibited a reciprocal regulation pattern

in response to 5-FC, suggesting that the process of 5-FC resistance is indeed controlled by Isw1, but in a disparate manner from other isolates. It is probable that the Isw1 regulation axis has undergone rewiring in these clinical isolates. However, it was shown that the CDLC135 strain overexpressing Isw1 exhibited sensitivity toward two azole drugs. The findings as mentioned above indicated that the modulation of Isw1 protein levels has a role in drug resistance observed in clinical isolates.

Taken together, our evidence demonstrates the critical function of Isw1 as a master regulator of multidrug responses in both laboratory and clinical strains. It allows us to decipher the molecular mechanism of the acetylation–Isw1–ubiquitination axis that modulates the expression of drug-resistant genes. These findings underscore the importance of performing thorough evaluations of PTMs in drug-resistance mechanism studies, highlighting a potential strategy for overcoming fungal drug resistance.

# Materials and methods

## Strains and growth conditions

Fungal cells (*Supplementary file 1*) were routinely grown in YPD medium (1% yeast extract, 2% peptone, and 2% dextrose). The biolistic transformation was performed using a YPD medium supplemented with 100 µg/ml nourseothricin (WERNER BioAgents), 200 µg/ml neomycin (Inalco), and 200 U/ml hygromycin B (Calbiochem) followed by colony selection. Deacetylases were blocked using 3 µM TSA (MedChemExpress) and 20 mM NAM (Sigma). Drug-resistance tests were performed using 16 µg/ml or 20 µg/ml FLC (MedChemExpress), 0.2 µg/ml or 0.3 µg/ml KTC (MedChemExpress), 100 µg/ml or 200 µg/ml 5-FC (MedChemExpress), 0.5 µg/ml or 1.0 µg/ml Amp B (MedChemExpress) or 50 µg/ml or 100 µg/ml 5-FU (MedChemExpress). The melanin synthesis was conducted by culturing it on L-DOPA agar medium at a temperature of 37°C. *C. neoformans* capsules were conducted in Dulbecco's modified Eagle medium, which was supplemented with 10% fetal bovine serum.

## Fungal proliferation in response to antifungal agents

The measurement of antifungal susceptibility was conducted using a dose–response method as previously described (*Iyer et al., 2023*; *Revie et al., 2022*), utilizing 96-well microtitre plates. In brief, dose–response tests were conducted using twofold serial dilutions of FLC (MedChemExpress), KTC (MedChemExpress), or 5-FC (MedChemExpress) in a final volume of 100 µl. Overnight YPD cultures were washed three times with phosphate-buffered saline (PBS) and diluted to an optical density of 0.02 at 600 nm. Following this, 100 µl of the resulting cell suspension was carefully dispensed into individual wells of a 96-well plate, with each well containing approximately 10,000 cells. The well plate was subjected to incubation at a temperature of 30°C for either 24 or 48 hr. Subsequently, optical density measurements at a wavelength of 600 nm were obtained using a Synergy HTX microplate reader manufactured by BioTek, as previously described (*Billmyre et al., 2020*). The growth of the relevant strain was standardized by normalizing it to the well without drug treatment. The average optical density values of technical duplicate measurements are depicted in the plot. Three biological replicates were conducted for each strain. The quantitative representation of dose–response assay results was visually shown using the software Java TreeView 1.1.3 (http://jtreeview.sourceforge.net) with a contrast value of 0.5. To facilitate the growth of dose–response assays, growth ratios were established for a minimum of three biological replicates. The data were graphed utilizing GraphPad Prism software.

## Chromatin immunoprecipitation and PCR

Co-IP of Isw1-Flag was performed as previously described (*Gao et al., 2022*). ISW1-FLAG was cultivated in YPD media overnight and subcultured to an $A_{600}$ of 0.8 (mid-log phase). 50 ml of the cell culture was added to 200 ml conical flasks containing 1% formaldehyde, and the flasks were incubated at room temperature with moderate rocking for 15 min. To end the cross-linking reaction, 2.5 M glycine was added to the mixture, which was then held for 5 min. At 4°C, cells were extracted at 1000 × *g* and washed twice with ice-cold PBS containing 125 mM glycine. After the chromatin is extracted, the DNA fragments are broken by ultrasound. The ultrasound condition is 3% power for 5 min (run for 3.5 s and then pause for 3 s). The fragment of the genome was then purified. IgG was used as the negative control for chromatin immunoprecipitation-qPCR. Quantitative real-time PCR (CFX96

real-time instrument; Bio-Rad) was used to analyze the gene abundance of immunoprecipitation using the specific primer pairs shown in Supplementary primers.

## Mass spectrometry

Mass spectrometry was performed to analyze Isw1 ubiquitination. The *ISW1-FLAG* complementation strains were subcultured at 30°C with 200 µM MG132 (MedChemExpress) in 50 ml YPD media at 30°C, and cells in the mid-log phase were used. Cell proteins were extracted using lysis buffer (50 mM Tris–HCl, 150 mM NaCl, 0.1% NP-40; pH 7.5) with 1× protease inhibitor cocktail (CWBIO) and 40 mM phenylmethylsulfonyl fluoride(PMSF). All lysed protein samples were incubated with anti-Flag magnetic beads (Sigma) at 4°C overnight. The beads were washed with Tris-Buffered Saline (TBS) buffer (50 mM Tris–HCl, 150 mM NaCl, 1% Triton X-100; pH 7.4) three times, and the bound proteins were extracted into protein loading buffer (125 mM Tris–HCl, 4% sodium dodecyl sulfate [SDS], 20% glycerol; pH 7.5) at 95°C for 5 min. All protein samples were separated using 8% SDS–polyacrylamide gel electrophoresis (PAGE), and the protein gel was stained with Coomassie Brilliant Blue R-250 (BBI Life Sciences) followed by clipping the gel strip.

In-gel tryptic digestion was performed by destaining the gel strips in 50 mM $NH_4HCO_3$ in 50% acetonitrile (vol/vol) until clear. The gel strips were dehydrated using 100 µl 100% acetonitrile for 5 min, rid of liquid, rehydrated in 10 mM dithiothreitol, then incubated at 56°C for 60 min. They were again dehydrated in 100% acetonitrile and rid of liquid, then rehydrated with 55 mM iodoacetamide followed by incubation at room temperature in the dark for 45 min. Next, they were washed with 50 mM $NH_4HCO_3$, dehydrated with 100% acetonitrile, then rehydrated with 10 ng/µl trypsin and resuspended in 50 mM $NH_4HCO_3$ on ice for 1 hr. After removing excess liquid, they were digested in trypsin at 37°C overnight, then peptides were extracted using 50% acetonitrile/5% formic acid followed by 100% acetonitrile.

The peptides were dried completely and then resuspended in 2% acetonitrile/0.1% formic acid. The tryptic peptides were dissolved in 0.1% formic acid (solvent A) then loaded directly onto a home-made reversed-phase analytical column (15 cm × 75 µm) on an EASY-nLC 1000 UPLC system. They were eluted at 400 nl/min using a gradient mobile phase that increased in solvent B (0.1% formic acid in 98% acetonitrile) from 6% to 23% over 16 min, from 23% to 35% over 8 min and from 35% to 80% over 3 min. Elution continued at 80% for an additional 3 min.

The peptides were subjected to an NSI source followed by tandem mass spectrometry (MS/MS) using a Q Exactive Plus (Thermo) mass spectrometer coupled to the UPLC. The electrospray voltage applied was 2.0 kV, the *m/z* scan range was 350–1800 for a full scan, and intact peptides were detected using an Orbitrap at a resolution of 70,000. Peptides were then selected for MS/MS using an NCE of 28, and the fragments were detected in the Orbitrap at a resolution of 17,500. The data-dependent procedure alternated between 1 MS scan and 20 MS/MS scans with a 15.0-s dynamic exclusion. The automatic gain control was set at 5E4.

The resulting MS/MS data were processed using Proteome Discoverer 1.3. Spectra were compared against acetylation, ubiquitination, or sumoylation databases. Trypsin/P (or other enzymes, if any) was specified as a cleavage enzyme, allowing up to four missing cleavages. Mass error was set to 10 ppm for precursor ions and 0.02 Da for fragment ions. Fixed modification was set to cysteine alkylation; variable modifications were set to lysine acetylation, ubiquitination, or sumoylation (QEQQTGG and QQQTGG), methionine oxidation, and protein N-terminal acetylation. Peptide confidence was set to 'high', and peptide ion score was set to '>20'.

## Strain generation

*C. neoformans* mutants were generated using the H99 strain and biolistic transformation (***Toffaletti et al., 1993***). The neomycin- or nourseothricin-resistance marker was amplified using primers M13F and M13R (***Supplementary file 3***). The upstream and downstream DNA sequences of the target gene and the selective marker sequence were joined using overlapping PCR. The resulting PCR fragments were purified, concentrated, and transformed into the H99 strain using biolistic transformation. Transformants were selected on YPD agar supplemented with either neomycin or nourseothricin. Correct integration and the loss of target DNA sequences were confirmed using diagnostic PCR.

The *ISW1* gene (*CNAG_05552*) was disrupted by the homologous replacement of its open reading frame (ORF) with a piece of DNA containing a dominant drug-resistance gene marker as described

henceforth. In the first round of PCR, the primer pairs 3534/3535 and 3536/3537 were used to amplify the 5′ and 3′ flanking regions, respectively, of the *ISW1* gene. The gel-extracted DNA fragments from the first round of PCR were used as templates, and the *isw1Δ::NEO* construct was amplified using the 3534/3537 primer pair. The H99 strain of *C. neoformans* was biolistically transformed with the deletion allele. To identify the desired *isw1Δ* mutant, diagnostic PCR was performed using the 3534/3537 primer pair, and real-time PCR followed, using primers 3557/3558. The same method was used to construct knockout strains of the E3 ligase-related genes in the *ISW1^{K97R}* strain. Briefly, the up- or downstream genomic DNA sequences of the target genes were amplified using the primers listed in the Supplementary Data primer table.

The *ISW1-FLAG* complementation strains were generated as described henceforth. The downstream genomic DNA sequence of *ISW1* was amplified using primers 3735/3736 and cloned between the restriction sites *Sac*II and *Sac*I into the pFlag-NAT plasmid (a plasmid containing the nourseothricin-resistance marker and the Flag tag). The complete ORF of the target gene (including the promoter sequence) was amplified using primers 3733/3734 and cloned between the restriction sites *Hind*III and *Eco*RI into pFlag-NAT-dw. The cassette amplified from the final pFlag-NAT by primer 3898-MY/3899-MY was biolistically transformed into the *isw1Δ::NEO* strain. Diagnostic PCR was performed using the 3733/3734 primer pair. Real-time PCR analysis was performed using a gene-specific probe amplified using the 3557/4421-MY primer pair. Western blot analysis of Isw1 was performed using anti-Flag mouse monoclonal antibodies.

The R mutation was formed using site-directed mutagenesis approaches. The K89 codon was first mutated using primers 3733/3866-MY and 3891/3734, then the K89R construct was amplified using the 3733/3734 primer pair and cloned between the restriction sites *Hind*III and *Eco*RI into pFlag-NAT-dw. The K97 codon was mutated using primers 3733/3787 and 3862-MY/3734, whereas the K113 codon was mutated using primers 3733/3836 and 3864-MY/3734. The resulting plasmids contained single-point mutants (K89R, K97R, or K113R), double-site mutants (K89R, K97R; K89R, K113R; or K97R, K113R) and triple-site mutant (K89R, K97R, and K113R). The Q mutation was formed using the same procedures except that the K89Q construct was formed using the primer pair 3733/3866-MY and 3892/3734, K97Q was formed using the primer pair 3733/3787 and 3863-MY/3734 and K113Q was formed using the primer pair 3733/3836 and 3865-MY/3734. The resulting plasmid contained single-point mutants (K89Q, K97Q, or K113Q), double-site mutants (K89Q, K97Q; K89Q, K113Q; or K97Q, K113Q), and triple-site mutant (K89Q, K97Q, and K113Q). All mutant plasmids were further confirmed using DNA sequencing. The cassette amplified from the final pFlag-NAT by primer 3898-MY/3899-MY was biolistically transformed into the *isw1Δ::NEO* strain. Mutant strains were confirmed using DNA sequencing, diagnostic PCR, qRT-PCR, and immunoblotting.

Ubiquitination mutants were also formed. The mutant K147R was formed using primer pair 3733/4445-MY and 4444-MY/3734, whereas K183R was amplified using the primer pair 3733/4447-MY and 4446-MY/3734 and K297R was mutated using primers 4448-MY/4449-MY. Similarly, K347R was mutated using primers 4450-MY/4451-MY, K415R was mutated using primers 3807-MY/3809-MY and K441R was mutated using primers 4452-MY/4453-MY. The resulting plasmids were used to generate the R mutant plasmid using the TaKaRaMutanBEST Kit (Takara). All strains were validated using the methods described earlier.

To demonstrate the direct protein interaction between Isw1 and Cdc4, the downstream genomic DNA sequence of *CDC4* was amplified using primers ZR34/ZR51 and cloned between the restriction sites *Spe*I and *Sac*I into the pHA-HYG plasmid (a plasmid containing the hygromycin B-resistance marker and the HA tag). Then, the last 1000 bp of the *CDC4* ORF was amplified using primers ZR48 and ZR33, and the resulting fragment was cloned into the above plasmid between *Cla*I and *Sma*I. The cassette amplified from the final plasmid by primers ZR48 and ZR51 was biolistically transformed into H99, the *ISW1^{WT}*, *ISW1^{K97Q}*, and *ISW1^{K97R}* strains. Diagnostic PCR was performed using the ZR48/ZR51 primer pair. Immunoblotting analysis of Cdc4 was performed using anti-HA (C29F4) rabbit mAb. The following strains were constructed using the same experimental procedure: Itc1-HA/Isw1-Flag, Chd1-HA/Isw1-Flag, and Sua7-HA/Isw1-Flag.

To detect the protein expression levels of Isw1 in clinical strains, the wild-type plasmid with pFlag-NAT was used as a template in PCR using primers 3557 and 3537. The resulting PCR products were transformed into seven clinical strains. The *ISW1^{K97Q}* overexpression strains were generated as described henceforth. A safe-haven site was applied to perform plasmid integration (***Arras et al.,***

*2015*). The 3′ flanking region of the safe haven was amplified using 4470-MY/4471-MY, then cloned into pFlag-NAT between the *Sac*II and *Sac*I sites. The 5′ flanking region of the safe haven was amplified using primer 4800-MY/4467-MY, the *TEF1* promoter was amplified using primer 4468-MY/2342 and the *ISW1^(K97Q)* coding sequence was amplified using 4807-MY and 3736. The *ISW1^(K97Q)* construct was amplified using the 4800-MY/3736 primer pair, and the three gel-extracted DNA fragments from the first and second rounds of PCR were used as templates, then cloned into pFlag-NAT-dw between the *Hind*III and *EcoR*I sites. The *ISW1^(K97Q)* overexpression cassette was amplified using the 4800-MY/4471-MY primer pair, and the product was transformed into clinical strains.

## Animal infection and *in vivo* drug-resistance tests

Mice were anesthetized and inoculated intranasally with $10^5$ yeast cells suspended in 50 µl PBS buffer. Infected mice were weighed 12 days after infection and then monitored twice daily for morbidity. Mice were sacrificed at the endpoint of the experiment. All animal experimentation was carried out under the approved protocol (please see Ethical statement).

The experimental design involved the implementation of *in vivo* investigations using a pre-established intranasal infection model as outlined by Oliveira NK (*Oliveira et al., 2021*). Each treatment group and control group consisted of 10 female BALB/c mice. Fungal cells were incubated for a period of 24 hr in a 10 ml volume of YPD medium at a temperature of 30°C. Subsequently, the cells were subjected to two rounds of washing and subsequently resuspended in PBS. All mice were subjected to an infection by intranasally introducing $10^5$ yeast cells. At 24 hr after infection, the mice were administered a single treatment with either FLC (at doses of 45 or 5 mg/kg) or 5-FC (at doses of 200 or 100 mg/kg) through intraperitoneal injection. The control group received a saline solution (PBS). The therapies were administered on a daily basis for a duration of 7 consecutive days. On the 14th day following infection, lung tissues were removed and homogenized, and then CFU analyses were performed.

## Protein co-IP mass spectrometry

Mid-log phase cells from ISW1-FLAG complementation strains subcultured at 30°C in 50 ml YPD medium were employed. Lysis buffer (50 mM Tris–HCl, 150 mM NaCl, 0.1% NP-40; pH 7.5) with 1× protease inhibitor cocktail (CWBIO) and 40 mM PMSF removed cell proteins. All lysate protein samples were treated overnight with Sigma anti-Flag magnetic beads at 4°C. After three TBS buffer washes, the bound proteins were extracted into protein loading buffer (125 mM Tris–HCl, 4% SDS, 20% glycerol; pH 7.5) at 95°C for 5 min. The protein solution was reduced with 5 mM dithiothreitol for 30 min at 56°C and alkylated with 11 mM iodoacetamide for 15 min at room temperature in darkness for digestion. Add 100 mM triethylammonium bicarbonate buffer (TEAB) to urea below 2 M to dilute the protein sample. Finally, trypsin was added at 1:50 for the first overnight digestion and 1:100 for the second 4 hr digestion. Finally, C18 SPE column desalted peptides.

The tryptic peptides were diluted in solvent A (0.1% formic acid, 2% acetonitrile/water) and put onto a homemade reversed-phase analytical column (25 cm length, 75/100 µm i.d.). A nanoElute UHPLC system (Bruker Daltonics) separated peptides with a gradient from 6% to 24% solvent B (0.1% formic acid in acetonitrile) over 70 min, 24% to 35% in 14 min, 80% in 3 min, and 80% for 3 min at 450 nl/min. Capillary source and timsTOF Pro (Bruker Daltonics) mass spectrometry were used on the peptides. The electrospray voltage was 1.60 kV. The TOF detector evaluated precursors and fragments with a 100–1700 *m/z* MS/MS scan range. PASEF mode was used on the timsTOF Pro. Precursors having charge states 0–5 were selected for fragmentation, and 10 PASEFMS/MS scans were obtained per cycle. The dynamic exclusion was 30 s.

MaxQuant search engine (v.1.6.15.0) handled MS/MS data. Tandem mass spectra were searched against the reverse decoy database and *C. neoformans* Protein-Fungi database (7429 entries). Up to two missed cleavages were allowed with trypsin/P. In first search, precursor ions had a mass tolerance of 20 ppm, in main search, 5 ppm, and fragment ions 0.02 Da. FDR was lowered to <1%.

All differentially expressed protein database accession or sequence was searched for protein–protein interactions in STRING 11.5. Only interactions between proteins in the searched dataset were chosen, avoiding extraneous candidates. STRING uses a 'confidence score' metric to measure interaction confidence. We identified interactions with a confidence value ≥0.15 (low confidence).

## Transcriptome and qRT-PCR analyses

To analyze drug resistance, the wild-type H99 and *isw1Δ* mutant strains were either untreated or were treated with 10 μg/ml FLC (Sigma) in 50 ml YPD media at 30°C until cell densities reached the exponential phase (approximately 6–7 hr). Cells were then washed three times with ice-cold PBS and placed in a tank of liquid nitrogen. Total RNA was isolated using TRIzol reagent (Thermo Fisher Scientific), and 3 μg of the product was processed using the TruSeq RNA Sample Preparation Kit (Illumina). Purification of mRNA was performed using polyT oligo-attached magnetic beads. Fragmentation of mRNA was performed using an Illumina proprietary fragmentation buffer. First-strand cDNA was synthesized using random hexamer primers and SuperScript II. Subsequently, second-strand cDNA was synthesized using RNase H and DNA polymerase I. The 3′ end of the cDNA sequence was adenylated, then cDNA sequences of 200 bp were purified using the AMPure XP system (Beckman Coulter) and enriched using an Illumina PCR Primer Cocktail in a 15-cycle PCR. The resulting PCR products were then purified, and integrity was confirmed using an Agilent High Sensitivity DNA assay on a Bioanalyzer 2100 (Agilent). The sequencing library was then sequenced using a Hiseq platform (Illumina) by Shanghai Personal Biotechnology Cp. Ltd. Alignments were checked against the *Cryptococcus_neoformans*_var._*grubii*_H99 reference genome and gene annotation set retrieved from Ensemble. Differentially expressed genes were detected using the Bioconductor package DESeq2 version 1.22.2 (*Gao et al., 2022*; *Love et al., 2014*). Genes with adjusted p-values <0.05 and changes greater or less than 1.5-fold those of the control strain were considered to be significantly induced or repressed, respectively.

To verify the gene changes screened by transcriptomics, H99 and *isw1Δ* mutant strains were either untreated or were treated with 40 μg/ml FLC (MedChemExpress) in 10 ml YPD media at 30°C, and cell densities were monitored until $OD_{600}$ reached 1.0. Both the H99 and *isw1Δ* mutant strains were either untreated or treated with 400 μg/ml 5-FC (MedChemExpress) in 10 ml YPD media at 30°C for 1 hr. Cells were harvested at 3000 rpm for 3 min at 4°C, then washed twice with ice-cold PBS. Total RNA was isolated using a total RNA kit I (Omega), and cDNA was synthesized using a reverse transcript all-in-one mix (Mona). Primers for amplifying target genes can be found in the primer table. Data were acquired using a CFX96 real-time system (Bio-Rad) using actin expression as a normalization control. The ΔΔCt method was used to calculate differences in expression.

## Co-IP and immunoblotting assays

Overnight cultures of *C. neoformans* strains were diluted in fresh YPD media and incubated at the indicated temperature to the mid-log phase ($OD_{600}$ = 0.8). Protein immunoprecipitation or co-IP was performed as described elsewhere (*Li et al., 2019*). Briefly, cell proteins were extracted using lysis buffer (50 mM Tris–HCl, 150 mM NaCl, 0.1% NP-40; pH 7.5) with 1× protease inhibitor cocktail (CWBIO) and 40 mM PMSF. Aliquots of protein extracts were retained as input samples. Samples of the lysed protein were incubated with anti-Flag magnetic beads (MedChemExpress) at 4°C overnight, then the beads were washed three times using TBS buffer, and the bound proteins were extracted into protein loading buffer at 95°C for 5 min. Protein samples were separated using 8% SDS–PAGE, transferred onto nitrocellulose membranes, and blocked using 5% milk. Immunoblotting or co-IP assays were performed using anti-Flag mouse monoclonal antibodies (1:5000 dilution; Transgene), anti-HA (C29F4) rabbit mAb (1:5000 dilution; Cell Signaling Technology), anti-Histone H3 (D1H2) XP Rabbit mAb (1:5000 dilution; Cell Signaling Technology), goat anti-mouse IgG (H+L) HRP secondary antibodies, and goat anti-rabbit IgG (H+L) HRP secondary antibodies (1:5000 dilution; Thermo Fisher Scientific), and monoclonal and polyclonal Kac (1:2500; PTM Bio). The mouse anti-acetyllysine primary antibody (clone Kac-10; PTM Bio, Cat No. PTM-101) was used to detect Kac (*Li et al., 2019*; *Xu et al., 2023*). The signal was captured using a ChemiDoc XRS+ (Bio-Rad). The resulting pictures were analyzed and quantified using Image Lab version 5.2.

## Statistical analysis

All statistical analyses were performed using GraphPad Prism software (GraphPad 6.0). Two-tailed unpaired *t*-tests were used in two-sample comparisons. Statistical analyses for two or more groups were performed using one- or two-way analysis of variance. Significant changes were recognized when p < 0.05. All experiments were performed using at least three biological replicates to ensure reproducibility.

## Detection of drug content

To analyze drug resistance, the wild-type H99 and *isw1Δ* mutant strains were treated with 40 μg/ml FLC in 50 ml YPD media at 30°C until the cell densities reached the exponential phase (approximately 5 hr), then the cells were washed once with PBS. An appropriate amount of uniform sample was weighed, 0.2 ml 50% acetic acid solution was added, ultrasonic extraction was carried out, and the resultant was passed through a 0.22-μm microporous filter membrane. High-performance liquid chromatography was performed using an injection volume of 10 nl and a constant mobile phase flow rate of 1.0 ml/min. An Agilent C18 (4.6 mm × 250 mm × 5 μm) column was used, held at 35°C, on a Thermo U3000 HPLC; the detector was a DAD. When the drug was FLC, the mobile phase was acetonitrile:water:acetic acid (25:75:0.2), the detector wavelength was 261 nm, the run time was 15 min, and the standard curve was $Y = 0.0147X - 0.0109$ ($r^2 = 0.9999$).

The drug content was determined as:

$$W = \frac{(C - C_0) * V * N}{m}$$

*W*—drug content, mg/kg
*C*—concentration of the drug in the cell, mg/l
$C_0$—concentration of the drug in the blank control, mg/l
*V*—volume, ml
*N*—diluted
*m*—cell mass, g

## Acknowledgements

We thank Profs. Yongqiang Fan and Ren Sheng for their critical review of the manuscript. This work was supported by the National Key Research and Development Program of China (2022YFC2303000). Funds for this program were also provided by the National Natural Science Foundation of China (31870140 to CD) and Liaoning Revitalization Talents Program (XLYC1807001 to CD). Research in PW lab was supported by the National Institutes of Health (US) awards AI156254 and AI168867.

## Additional information

### Funding

| Funder | Grant reference number | Author |
| --- | --- | --- |
| National Key Research and Development Program of China | 2022YFC2303000 | Chen Ding |
| National Natural Science Foundation of China | 31870140 | Chen Ding |
| Liaoning Revitalization Talents Program | XLYC1807001 | Chen Ding |
| National Institutes of Health | AI156254 | Ping Wang |
| National Institutes of Health | AI168867 | Ping Wang |

The funders had no role in study design, data collection, and interpretation, or the decision to submit the work for publication.

### Author contributions

Yang Meng, Conceptualization, Data curation, Formal analysis, Validation, Investigation, Visualization, Methodology, Writing - original draft; Yue Ni, Data curation, Formal analysis, Methodology; Zhuoran Li, Data curation, Investigation, Methodology; Tianhang Jiang, Data curation, Methodology; Tianshu Sun, Conceptualization, Project administration; Yanjian Li, Conceptualization, Methodology; Xindi

Gao, Software, Visualization, Methodology; Hailong Li, Software, Funding acquisition, Methodology; Chenhao Suo, Chao Li, Sheng Yang, Methodology; Tian Lan, Software; Guojian Liao, Tongbao Liu, Resources, Writing - review and editing; Ping Wang, Funding acquisition, Writing - review and editing; Chen Ding, Conceptualization, Resources, Supervision, Funding acquisition, Methodology, Writing - original draft, Project administration, Writing - review and editing

**Author ORCIDs**
Chen Ding  http://orcid.org/0000-0002-9195-2255

## Ethics

All animal experiments were reviewed and ethically approved by the Research Ethics Committees of the National Clinical Research Center for Laboratory Medicine of the First Affiliated Hospital of China Medical University (KT2022284) and were carried out in accordance with the regulations in the Guide for the Care and Use of Laboratory Animals issued by the Ministry of Science and Technology of the People's Republic of China. Infections with C. neoformans were performed via the intranasal route. Four- to six-week-old female Balb/c mice were purchased from Changsheng Biotech (Liaoning, China) and used for survival and fungal burden analyses.

## Decision letter and Author response

Decision letter https://doi.org/10.7554/eLife.85728.sa1
Author response https://doi.org/10.7554/eLife.85728.sa2

# Additional files

## Supplementary files

- Supplementary file 1. Strains used in this study.
- Supplementary file 2. DEGs (differentially expressed genes) in *isw1Δ* cells treated with or without fluconazole (FLC).
- Supplementary file 3. Primers used in this study.
- Supplementary file 4. Co-immunoprecipitation (Co-IP) mass spectrometry analysis of Isw1-Flag.
- Supplementary file 5. Comparison of drug-resistant genes among clinical isolates.
- MDAR checklist

## Data availability

The raw Isw1 proteome modification mass spectrometric data have been deposited to the Proteome Xchange with identifier PXD037150. The mass spectrometry proteomics data have been deposited to the ProteomeXchange Consortium via the iProX partner repository (*Chen et al., 2022*) with the dataset identifier PXD045338. The transcriptomics data (RNA-seq) is deposited in NCBI's Gene Expression Omnibus (GEO) and can be accessed through GEO Series accession ID GEO:GSE217187 and GSE235148. Any other data necessary to support the conclusions of this study are available in the supplementary data files and source data. Reagents and fungal strains are available from the authors upon request.

The following datasets were generated:

| Author(s) | Year | Dataset title | Dataset URL | Database and Identifier |
|---|---|---|---|---|
| Ding C | 2022 | Interplay between acetylation and ubiquitination of imitation switch chromatin remodeler Isw1 confers multidrug resistance in Cryptococcus neoformans | http://www.ncbi.nlm.nih.gov/geo/query/acc.cgi?acc=GSE217187 | NCBI Gene Expression Omnibus, GSE217187 |

*Continued on next page*

*Continued*

| Author(s) | Year | Dataset title | Dataset URL | Database and Identifier |
|---|---|---|---|---|
| Yang M | 2022 | Interplay between acetylation and ubiquitination of imitation switch chromatin remodeler Isw1 confers multidrug resistance in Cryptococcus neoformans | https://www.ebi.ac.uk/pride/archive/projects/PXD037150 | PRIDE, PXD037150 |
| Yang M | 2023 | Interplay between acetylation and ubiquitination of imitation switch chromatin remodeler Isw1 confers multidrug resistance in Cryptococcus neoformans | https://proteomecentral.proteomexchange.org/cgi/GetDataset?ID=PXD045338 | ProteomeXchange, PXD045338 |
| Ding C | 2023 | Interplay between acetylation and ubiquitination of imitation switch chromatin remodeler Isw1 confers multidrug resistance in Cryptococcus neoformans | http://www.ncbi.nlm.nih.gov/geo/query/acc.cgi?acc=GSE235148 | NCBI Gene Expression Omnibus, GSE235148 |

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
