## [Editor Report]

This important study makes a solid connection between chromatin remodeling, post-translational regulation, and antifungal drug resistance in Cryptococcus neoformans, revealing a new facet of how drug resistance can emerge. Establishing a link between chromatin remodeling and antifungal resistance is a finding that will be of interest to infectious disease researchers, cell biologists, and drug developers.

---

## [Decision Letter]

**Decision letter after peer review:**

Thank you for submitting your article "Interplay between acetylation and ubiquitination of imitation switch chromatin Cryptococcus neoformans" for consideration by *eLife*. Your article has been reviewed by 3 peer reviewers, and the evaluation has been overseen by a Reviewing Editor and Arturo Casadevall as the Senior Editor. The reviewers have opted to remain anonymous.

Essential revisions:

We would welcome a revision that addresses all the reviewer points below. In addition please pay particular attention to these comments that emerged during the reviewer discussion of your paper.

1) it is important to more firmly demonstrate whether the Isw1 effect is due to its chromatin activity.

2) Isw1 is not well characterised in C. neoformans. It is important to demonstrate that Isw1-protein complexes are conserved between different organisms- the authors should have this info thanks to their Mass-Spec data.

3) Can you explain how Iswi was found to be important in antifungal resistance? Are Iswi paralogs present in Cryptococcus? What potential Iswi interacting proteins are present in Cryptococcus other than Itc1?

4) Can you explain the lack of high molecular-weight ubiquitinated Isw1 proteins (I am not an expert in ubiquitination but I thought that poly-ubiquitination always produced a different ladder) and

5) The specificity of the KAc antibody used needs to be better defined.

6) The authors should calculate the MIC using standard methods.

The revised manuscript will be sent back to the reviewers for re-review so please make sure you answer all the points above and below if you resubmit to *eLife*.

*Reviewer #1 (Recommendations for the authors):*

1. It would greatly enhance the paper if the authors could show that the isw1 mutant is resistant to antifungal treatment in mouse models. Given that the mutant is fully virulent, the difference in survival or fungal burden after drug treatment would indicate the potential role of Isw1 during antifungal therapy. That would enhance the impact of the research showing the potential clinical relevance of Isw1 regulation.

2. The MICs shown in Figure 1b and Figure 6f were not done according to standard protocols. Although the spotting assays are nice because they are visual, the authors need to use the accepted standard protocol to determine MICs of antifungals for both the mutants and the clinical isolates.

3. It's surprising that all the clinical isolates used in this study showed altered Isw1 levels. The authors could include clinical isolates with known mechanisms of resistance for comparison.

4. The authors believe that reduction in 5-FC uptake contributes to the resistance of the isw1 mutant to this drug. Given that isw1 mutation altered the expression of many genes (but none of the changes of the transporter genes are huge as the log2 FPKM is within 1.5, Figure 1C) and that there is only one known transporter Fcy2 involved in the intake of 5-FC, this presents an ideal case for the authors to demonstrate that reduced uptake is indeed the underlying resistance mechanism.

*Reviewer #2 (Recommendations for the authors):*

General comments:

The term protein posttranslational modification (PPTM) is more commonly referred to as post-translational modification (PTM). The authors use PPTM throughout this manuscript. The authors could consider changing "PPTM" to "PTM".

The authors' use of "increase", "induce", or similar wording is confusing in some instances. For example, in Lines 197-198 the authors suggest that deacetylase inhibitors "increase acetylation levels" (see also Lines 235, 283, and 284 for additional examples). The authors should consider rewording these examples and reviewing the manuscript for similar examples to clarify their results.

The authors find that deletion of ITC1, which is known to interact with Isw1 homologs in other systems, results in increased antifungal resistance. However, loss of Itc1 does not fully phenocopy loss of Isw1 (see Figure 1—figure supplement 2 panel B). This is likely due to the activity of other Isw1 interacting proteins. The authors are encouraged to review recent literature on other organisms (particularly other fungi, such as *Neurospora crassa*) on other potential interacting proteins.

The authors found that deletion of CDC4 in the Isw1 K97R background results in reduced resistance to antifungals. It also appears that the deletion of CDC4 in the Isw1 K97R background impacts general growth as shown on YPD in Figure 5 (panel A). Did the authors examine the deletion of CDC4 in the wild-type background? How does deletion of CDC4 in wild type impact general growth and resistance to antifungals?

The authors state that "results showed an interaction between Cdc4 and Isw1 K97R but

not between Cdc4 and Isw1 K97Q" (Line 269) in reference to data shown in Figure 5 (panel E). However, it appears that Cdc4 and Isw1 K97Q show a very weak interaction based on the co-IP data. The authors should address this discrepancy.

The authors are encouraged to thoroughly review and revise the "Material and Methods" section and figure legends of the manuscript to ensure methods are described in sufficient detail to repeat, citations are provided when needed, and figures are adequately described. For example: (1) although the manuscript relies heavily on the quantification of western blot data the authors do not describe how this quantification was performed, and (2) the authors do not describe how RNA-seq data was processed or mapped (see Line 559) or provide a citation for DESeq2 package (see Line 562).

It is not clear the number of biological replicates used or if experiments were repeated for some experiments. For example: (1) in Figure 3 panels A and B and related source data, it appears these experiments were run once with one biological replicate, and (2) in Figure 3 panel F and related source data, it appears that western blots for two replicates are shown, although the data for three replicates are plotted. Could the authors clarify the number of replicates used and if experiments were repeated?

Specific comments:

Line 68: The authors should rewrite/edit Line 68 for clarity.

Line 69: The citation, "(2017)", is incomplete. The authors should correct the citation here and check throughout the manuscript for additional instances of incomplete citations (e.g., Line 74).

Line 103: It appears the Florent et al., 2009 citation is incorrectly used here, as the referenced study investigated antifungal resistance in Candida lusitaniae.

Line 134: The authors could consider briefly describing why they chose to study Isw1 in Cryptococcus.

Line 136: The authors should specify that the isw1 mutant was complemented with an ISW1 allele tagged with FLAG.

Line 154: The authors state "To test if this increase affects drug uptake,…"; however, the experiment described doesn't distinguish between decreased drug import and increased drug export. This should be rewritten for clarity.

Line 180: The authors should rewrite/edit Line 180 for clarity.

Line 191: The authors should rewrite/edit Line 191 for clarity.

Line 200: The authors should rewrite/edit Line 200 for clarity and define the abbreviation for DNA binding domain as only "DBD" is used in Figure 2 (panel F) to avoid confusion. Additionally, the authors should describe why certain acetylated residues were chosen and others were not.

Lines 201-202: The authors should revise the statement "to mimic a fully acetylated Isw1" for clarity and correctness, as there appear to be additional acetylated lysine residues in Isw1.

Lines 208-209: The authors should rewrite/edit Lines 208-209 for clarity.

Lines 220-221: The authors should rewrite/edit Lines 220- a221 for clarity.

Line 246: Data presented in this study suggest 15 residues of Isw1 are ubiquitinated but only six were mutated. The authors should provide reasoning for selecting certain ubiquitination sites in the manuscript. Specifically, it is curious that the authors did not test K98 as the proximity to K97 could affect acetylation or vice versa. Can the authors explain why K98 was not tested? Additionally, the authors state that "five failed to affect drug-resistant growth phenotypes". However, it seems that two of the six tested residues had an impact on drug resistance in the K97R background. The authors should address this discrepancy.

Lines 250-252: The authors should rewrite/edit Lines 250-252 for clarity.

Line 277: "majority was" should be changed to "majority were" in this instance.

Lines 307-308: Gcn5 is not a deacetylase but is the catalytic subunit of the histone acetyltransferase complexes, ADA, and SAGA. The authors should address this discrepancy.

Line 313: The authors use "the acetylation-Isw1-ubiquitination axis" here and use the "Isw1 acetylation-ubiquitin-proteasome regulation axis" above (see Line 289). The authors could consider using consistent wording throughout the manuscript for clarity.

Line 315: It appears the authors use a lowercase "L" instead of an uppercase "I" for Ioc3 and Ioc4. The authors should address this discrepancy.

Lines 320-323: The authors do not directly demonstrate that transporter-encoding genes upregulated in the ISW1 deletion strain are responsible for reduced FLC in cells or if the reduction of FLC in cells is caused by decreased drug import and/or increased drug export (see comment for Line 154 above). Relatedly, it should be noted that Figure 1 (panel C) indicates multiple transporter-encoding genes are downregulated in the ISW1 deletion strain, and it is not clear if the authors describe and/or discuss this in the text. With these points in mind, the authors should rewrite Lines 320-323 for clarity.

Lines 324-327: The authors should rewrite/edit Lines 324-327 for correctness and clarity. For example, the authors state "Isw1 is a transcription activator for genes responsible for resistance to FLC"; however, it seems that genes involved in FLC resistance are upregulated in the absence of Isw1, which means Isw1 activity represses expression of such genes. The authors are also encouraged to keep in mind that Isw1 is an ATP-dependent chromatin remodeling factor and not a transcription factor.

Figure 2 panels B and D: The authors should define CHX in the figure legend, i.e., "cycloheximide (CHX)".

Figure 4 panel E: It appears the images showing growth on 16 ug/ml FLC and 20 ug/ml FLC may be mislabeled. The authors should review and correct if needed.

*Reviewer #3 (Recommendations for the authors):*

A) How did the author identify ISWI as a candidate gene important for regulating drug resistance in Cryptococcus?

B) The budding yeast, *S. cerevisiae*, expresses two homologs of ISWI, Isw1p, and Isw2p. Isw2p associates with Itc1p (Sugiyama and Nikawa, 2001) The Isw1p forms two distinct complexes: one subcomplex is formed by Isw1p, Ioc2p, and Ioc4p and a second by Isw1p and Ioc3 (Vary Mol Cell Biol 2003). Does Cryptococcus contain two ISWI genes? If so, why did the authors focus just on the Isw(2?)-Itc1p proteins? Does Cryptococcus encode for Ioc2, Ioc3, and Ioc4? The authors should clarify this point.

C) The itc1 deletion strain is barely resistant to FLC (Figure S2): MIC for isw1 δ and itc1 δ strains should be calculated.

D) Did the authors perform transcriptomic analysis of WT and isw1 deleted cells without FLC? The material and methods section suggests that this experiment has been performed (Pg 18 line 544). However, these RNA-seq data are not shown in Figure 1. This is a critical experiment as it establishes whether ISWI is an activator or a repressor of gene expression.

E) Isw1-FLAG pulldown- It is strange that the authors dive into Isw1 post-translational modifications without describing the pull-down and characterising the protein complexes. What is the efficiency of the pull-down? This could be shown by Silver staining+ Western blot. What are the Iswi1-interacting proteins? Are the ISWI-protein complexes conserved in Cryptococcus?

F) For all the FLAG-Western Blots: No FLAG control is missing.

G) For all the FLAG Pulldowns: Input is missing.

H) What does the anti-Kac detect? Iswi1 acetylation? Did the authors raise the antibody? Histone acetylation? I cannot find this info in the Material and Methods. Assuming that the antibody detects Isw1 acetylation, I am not convinced that "the presence of antifungal agents strongly repressed acetylation levels" (Line 196) as reduced protein acetylation levels are observed only when FLC treatment is combined with TSA/NAM treatment.

I) Although the mutation analysis demonstrates that K97 plays a role in the drug response- data demonstrating that acetylation of this residue regulates this process are missing. Could the mutation affect ISWI folding? The crystal structure of several ISWI proteins has been solved and it should be relatively easy to predict the effect of K97 mutations on ISWI protein folding.

J) Protein poly-ubiquitination (normally linked to proteosome-degradation) leads to higher molecular weight protein species in a Western blot analysis. Therefore, it is puzzling that these high molecular weight species are not detected, especially after MG132 treatment.

K) Is the Cdc4-Iswi interaction detected by Mass Spec Analysis?

L) Is ISWI function in drug resistance linked to its chromatin remodelling activity?

M) Does ISWI interact with chromatin? If so, which are ISWI-target genes? Does drug treatment modulate chromatin binding?

[Editors' note: further revisions were suggested prior to acceptance, as described below.]

Thank you for resubmitting your work entitled "Interplay between acetylation and ubiquitination of imitation switch chromatin remodeler Isw1 confers multidrug resistance in *Cryptococcus neoformans*" for further consideration by *eLife*. Your revised article has been evaluated by Detlef Weigel (Senior Editor) and a Reviewing Editor.

While the manuscript has been improved there are some remaining issues that need to be addressed, as outlined below:

*Reviewer #2 (Recommendations for the authors):*

General comments on revision: The revised manuscript is much improved overall. The authors have performed numerous additional experiments that strengthen the study. The majority of my initial comments have been addressed, and I thank the authors for carefully considering the comments and their responses. However, there are a few key points in the manuscript (largely associated with revisions or new data) that should be addressed to further strengthen the study and broaden its impact.

Prior reviewer comment: The authors demonstrate that Isw1 has a role in responding to antifungals in Cryptococcus. However, it is not clear if changes in Isw1 stability represent a general response to stress. This study would have benefited from experiments to test: (1) if levels of Isw1 change in response to other stressors (e.g., heat, osmotic, or oxidative stress) and (2) if loss of Isw1 impacts resistance to other stressors.

Author response: A series of experiments were conducted to illustrate and measure phenotypic traits associated with virulence. These traits encompassed capsule formation, melanin synthesis, cell proliferation under stressful conditions, and Isw1 expression levels in response to diverse environmental stimuli. Please see Figure 3a, 3b, 3c, Figure 3—figure supplement 1 and line 237-241.

Reviewer response: The authors provide new data to address the initial comment. The phenotypic data presented in Figure 3a, 3b, and 3c are convincing, although it should be noted that there appears to be a slight reduction in melanin production in Figure 3b. The data presented in Figure 3 —figure supplement 1 is less convincing. It is likely difficult to say "that Isw1 expression is not affected by external stress inducers" (line 237) considering data presented in Figure 1 —figure supplement 1 and Figure 4. Specifically, (1) ISW1 expression is not affected by antifungal treatment under the conditions tested and (2) changes in Isw1 levels associated with antifungal treatment are observed after inhibiting protein synthesis. It is thus plausible that changes in Isw1 levels under stress conditions could be missed in the assay presented in Figure 3 —figure supplement 1. The authors should change the text accordingly.

Prior reviewer comment: The authors find that deletion of ITC1, which is known to interact with Isw1 homologs in other systems, results in increased antifungal resistance. However, loss of Itc1 does not fully phenocopy loss of Isw1 (see Figure 1 —figure supplement 2 panel B). This is likely due to the activity of other Isw1 interacting proteins. The authors are encouraged to review recent literature on other organisms (particularly other fungi, such as *Neurospora crassa*) on other potential interacting proteins.

Author response: We acknowledge the significance of the conservation of Isw1 across many species. The regulatory pathway of Isw1 was initially discovered in the model organism *Saccharomyces cerevisiae*, which possesses two paralogs, namely Isw1 and Isw2, as a result of a whole genome duplication event(Kellis M, 2004; Tsukiyama T, 1999; Wolfe KH, 1997). Due to the absence of a complete genome duplication event in C. neoformans, its genome solely contains a single copy of the ISW gene. Prior research conducted on *Saccharomyces cerevisiae* has provided evidence that the ISWI complex is composed of several subunits, namely Isw1, Ioc proteins, Itc1, Chd1, and Sua7 (Mellor J, 2004; Smolle M, 2012; Sugiyama and Nikawa, 2001; Vary JC Jr, 2003; Yadon AN, 2013). The genome of C. neoformans was examined, and it was observed that the IOC gene family could not be identified. This suggests that the IOC gene family has likely undergone an evolutionary loss in C. neoformans, as indicated on the FungiDB website. The genome of C. neoformans harbors three distinct genes, namely Itc1, Chd1, and Sua7. In order to comprehensively investigate the cryptoccocal ISWI complex, we conducted a methodical Isw1-Flag protein immunoprecipitation procedure, which was subsequently followed by Mass-Spec analysis. In the present study, a total of 22 proteins that interacted with Isw1 were discovered. Among these proteins, 11 have been previously reported to be associated with the regulatory networks including Isw1. In the mass spectrometry results, Itc1 was found to be co-immunoprecipitated with Isw1. Although the Mass-Spec analysis did not reveal the presence of Chd1 and Sua7, our study demonstrated that Chd1 can be coimmunoprecipitated with Isw1 through co-IP and immunoblotting techniques. However, no interaction between Isw1 and Sua7 was established utilizing any of these methods. In order to gain a deeper understanding of the involvement of Chd1 and Itc1 in the regulation mechanism of Isw1 in multidrug resistance, we created strains with disrupted chd1Δ and itc1Δ genes. The only strain that exhibited a drug resistance phenotype similar to that of the isw1Δ strain was the itc1Δ strain. The data presented in this study indicate that there has been evolutionary divergence in the Isw1-protein complexes between the species C. neoformans and *S. cerevisiae*. The manuscript has undergone significant modifications. Please see Figure 2 and text line 208-232.

Reviewer response: The authors provide new data to data address the initial comment. This data greatly improves the manuscript. However, it also raises multiple concerns that must be addressed.

1. The authors should be mindful that (1) there is not a single "ISWI Complex" in *S. cerevisiae* or other eukaryotes, but rather ISWI homologs form multiple complexes (e.g., Isw1a, Isw1b, and Isw2 in *S. cerevisiae*); and (2) the IOC (Iswi One Complex) genes encoding Ioc2, Ioc3, and Ioc4 are not a "gene family" per se as the proteins are quite different, and it may be difficult to readily identify divergent homologs in C. neoformans. Additionally, the authors should clarify what is meant by "canonical ISWI complex" (line 231).

2. It is unclear why the authors chose to focus on Chd1 and Sua7 as (1) neither Chd1 nor Sua7 appear in the AP-MS data and (2) there is very little evidence that Chd1 or Sua7 directly interact with Isw1 or Isw2 in *S. cerevisiae* (to the best of my knowledge, Michaelis et al., 2023 Nature provides the only biochemical evidence for interaction between Isw1 and Chd1 in *S. cerevisiae*). There is evidence that both Chd1 and Sua7 genetically interact with Isw1 or Isw2 in *S. cerevisiae*, which is what is described in the references cited by the authors (for example, Smolle et al., 2012). Along these lines, the authors should review the text and references and revise both accordingly. It should be noted that the evidence for an association between Isw1 and Chd1 in C. neoformans is quite interesting (although it may not be a direct physical interaction).

3. As mentioned in the initial comment, the authors are encouraged to review recent literature on ISWI homologs in organisms other than *S. cerevisiae* (e.g., *Neurospora crassa* – Kamei et al., 2021 PNAS and Wiles et al., 2022 *eLife*) to strengthen this aspect of the manuscript. For example, there are at least two additional proteins in the AP-MS data that are well-characterized to interact with ISWI homologs. Along these lines, the authors should provide protein names in addition to locus IDs in Figure 2a (as was done for Itc1) to help readers and broaden interest.

Prior reviewer comment: The authors' use of "increase", "induce", or similar wording is confusing in some instances. For example, in Lines 197-198 the authors suggest that deacetylase inhibitors "increase acetylation levels" (see also Lines 235, 283, and 284 for additional examples). The authors should consider rewording these examples and reviewing the manuscript for similar examples to clarify their results.

Author response: The manuscript has been appropriately updated in accordance with the given instructions.

Reviewer response: Many such instances were not corrected (for example, see lines 281 and 322-323). Similarly, (1) in lines 127-128, the authors state "Cryptococcal Isw1 plays an indispensable role in modulating drug-resistance genes", but the role of Isw1 is not indispensable since the isw1 mutant strain has increased resistance to antifungals; and (2) in line 247, the authors state "that the isw1 mutant strain exhibits improved fungal burdens", but the isw1 mutant strain showed increased fungal burdens not improved fungal burdens.

---

## [Author Response]

Essential revisions:We would welcome a revision that addresses all the reviewer points below. In addition please pay particular attention to these comments that emerged during the reviewer discussion of your paper.1) it is important to more firmly demonstrate whether the Isw1 effect is due to its chromatin activity.

In order to investigate the potential role of Isw1 on chromatin activity in the modulation of multidrug resistance, we have conducted protein truncation experiments. Specifically, we deleted the DNA binding domain, the helicase domain, and the SNF2 domain, which have been previously shown to regulate Isw1 chromatin activity in the model organism *S. cerevisiae* (Grune T, 2003; Mellor J, 2004; Pinskaya M, 2009; Rowbotham SP, 2011). The new data demonstrated that all truncation variants of Isw1 mutants had a growth phenotype consistent with that of the deletional strain *isw1Δ*. In addition, the levels of gene expression observed in these strains were also similar to those observed in the deletion strain *isw1Δ*. This finding provides evidence that the regulation of the drug resistance mechanism is influenced by these critical domains involved in modifying chromatin activities. Moreover, the Isw1-Flag strain was utilized to conduct chromatin immunoprecipitation and PCR experiments, which revealed that Isw-1 exhibits the ability to directly bind to the promoter regions of target genes. The new findings added evidence substantially supporting the hypothesis that the Isw1 chromatin activity plays a crucial role in modulating its protein function and acting as a central regulator of drug resistance in *C. neoformans*.

Please see revised Figure 1g, 1h, 1i, and lines 186-199 in the revised manuscript text.

2) Isw1 is not well characterised in C. neoformans. It is important to demonstrate that Isw1-protein complexes are conserved between different organisms- the authors should have this info thanks to their Mass-Spec data.

We acknowledge the significant concern raised over the conservation of Isw1 across several species. The regulatory mechanism of Isw1 was initially discovered in *S. cerevisiae*. This process involves two paralogs, Isw1 and Isw2, which emerged as a result of the complete genome duplication event ((Kellis M, 2004; Tsukiyama T, 1999; Wolfe KH, 1997)). Because no evidence suggested that *C. neoformans* has gone through similar genome duplication, only one copy of ISW gene was identified. Previous research in *S. cerevisiae* has provided evidence that the ISWI complex is comprised of several subunits, namely Isw1, Ioc, Itc1, Chd1, and Sua7 (Mellor J, 2004; Smolle M, 2012; Sugiyama and Nikawa, 2001; Vary JC Jr, 2003; Yadon AN, 2013). Upon a thorough examination of the *C. neoformans* genome, we have not been able to identify a similar *IOC* gene family. This absence likely suggests an evolutionary loss of the *IOC* gene family in *C. neoformans*, as reported on the FungiDB website (https://fungidb.org/fungidb/app). However, *C. neoformans* has Itc1, Chd1, and Sua7. While we concur with the aforementioned statement on the capability of Mass-Spec data to elucidate potential protein-protein interactions and aid in the identification of subunits within the ISWI complex, it is important to acknowledge that the PTM Mass-Spec methodology is solely employed for the purpose of identifying potential sites of protein modification. In order to comprehensively investigate the cryptoccocal ISWI complex, we conducted a standardized Isw1-Flag protein immunoprecipitation procedure, followed by Mass-Spec analysis. In the present study, a total of 22 proteins that interacted with Isw1 were found. Among these proteins, 11 have been previously reported to be associated with the regulatory networks including Isw1. In the mass spectrometry results, Itc1 was found to be co-immunoprecipitated with Isw1. Although the Mass-Spec analysis did not reveal the presence of Chd1 and Sua7, our study demonstrated that Chd1 can be coimmunoprecipitated with Isw1 through co-IP and immunoblotting techniques. However, no interaction between Isw1 and Sua7 was established utilizing any of these methods. In order to gain a deeper understanding of the involvement of Chd1 and Itc1 in the regulation mechanism of Isw1 in multidrug resistance, we created *chd1Δ* and *itc1Δ* disruption mutant strains. The only strain that exhibited a drug resistance phenotype similar to the *isw1Δ* strain is the *itc1Δ* strain. The new data indicated that there is evolutionary divergence in the Isw1-protein complexes between *C. neoformans* and *S. cerevisiae*. We have added new findings and revised relevant statements. Please see new Figure 2, Supplementary File 4, and text lines 206–232.

3) Can you explain how Iswi was found to be important in antifungal resistance? Are Iswi paralogs present in Cryptococcus? What potential Iswi interacting proteins are present in Cryptococcus other than Itc1?

Isw1 was identified in a further investigation built upon the findings presented in our previously published studies (Li Y, 2019). In this study, the acetylome in *C. neoformans* was comprehensively analyzed, and a series of knockout strains were created to investigate the relationship between acetylation and fungal pathogenicity. The *isw1* mutant was discovered to be a modifier of drug resistance. The identification of fungal paralogs of *ISW* genes was initially observed in *S. cerevisiae*, which has two paralogs, Isw1 and Isw2, as a result of genome duplication (Kellis M, 2004; Tsukiyama T, 1999; Wolfe KH, 1997). In contrast, *C. neoformans* has only one *ISW1* gene. We have conducted bioinformatic analysis, protein co-immunoprecipitation followed by Mass Spectrometry, immunoblotting experiments, and gene disruption studies to characterize Isw1. In addition, we found that *C. neoformans* does not have all of the IOC proteins as *S. cerevisiae* does and the cryptococcal Isw1-protein associated complex exhibits substantial divergence from the yeast. We established connections between Isw1-Itc1 and -Chd1, but not Isw1-Sua7, in *C. neoformans*. Consequently, we have conducted additional investigations to also elucidate the role of Chd1 and Itc1 in regulating drug resistance. It is noteworthy that among the studied strains, only Itc1 exhibited a reciprocal regulation of drug resistance in *C. neoformans*. This particular phenotype bears resemblance to that seen in the cryptoccocal *isw1Δ* strain.

Please see revised Figure 2, Supplementary File 4, and text lines 206–232.

4) Can you explain the lack of high molecular-weight ubiquitinated Isw1 proteins (I am not an expert in ubiquitination but I thought that poly-ubiquitination always produced a different ladder) and

In theory, the process of poly-ubiquitination results in the formation of high molecular-weight proteins that exhibit smear patterns. However, when employing the immunoblotting technique to analyze poly-ubiquitination proteins, it is common to observe the detection of the apo form of the target proteins (i.e., non-ubiquitinated forms of the protein). This is mostly due to the low abundance of ubiquitinated protein species. It is also crucial to note that the presence of a smear in a detection assay does not provide definitive evidence of ubiquitination, as the smear may be attributed to various other factors. Hence, the identification of ubiquitinated proteins necessitates an additional enrichment step, such as the protein immunoprecipitation (IP) assay, followed by the detection of ubiquitin through the utilization of anti-ubiquitin antibodies. In addition, it should be noted that mass spectrometry is considered the most precise technique for the detection of ubiquitination. It is worth mentioning that our original manuscript included data obtained using MS analysis. In order to provide a more precise account of conventional techniques employed for the detection of ubiquitination using immunoblotting, we presented below two illustrative instances of ubiquitination detection investigations that have been recently published. Both experimental protocols involved IP, followed by immunoblotting analysis employing anti-ubiquitin (UB) antibodies.

Study 1. Feeding induces cholesterol biosynthesis via the mTORC1–USP20–HMGCR axis. 2019, Nature, https://doi.org/10.1038/s41586-020-2928-y

Please see Figure 1 for HMGCR protein IP followed by Ub detection using the Ub antibody

Study 2. USP10 strikes down b-catenin by dual-wielding deubiquitinase activity and phase separation potential, Cell Chemical Biology. https://doi.org/10.1016/j.chembiol.2023.07.016

Please see Figure E,F,H for Axin1 protein IP followed by Ub detection using the Ub antibody

In our studied, we have seen that the native form of Isw1 can be detected using immunoblotting, but the presence of high molecular weight protein species remains undetectable. This finding aligns with the outcomes reported in the aforementioned papers. In the revised manuscript, an attempt was made to investigate the impact of IP Isw1 and measure ubiquitination levels using commercially available mammalian Ub antibodies. However, these efforts did not yield successful results. In order to address the concerns raised, we successfully engineered a strain of *C. neoformans* capable of producing the human UBB1 protein. Experiments using ubiquitination were conducted. Three strains were subjected to testing, namely the wild-type strain, the ISW1K97Q strain, and the ISW1K97R strain. The cells were subjected to incubation in media that were either supplemented with MG132 or not. The experimental protocol involved protein pull down utilizing Flag beads, followed by the identification and quantification of ubiquitin with anti-human ubiquitin antibodies. Following an extended period of exposure, we successfully observed the presence of poly-ubiquitinated protein smear bands, which corresponded to the elevated molecular weight of poly-ubiquitinated Isw1. The results revealed that the presence of poly-ubiquitinated Isw1 species exclusively in the ISW1K97R strain, while not in the wild-type and ISW1K97Q strains. The presence of the K97Q mutant inhibits the ubiquitination of Isw1. These new data provided additional evidence to support the hypothesis that Isw1 is a protein undergoing ubiquitination. Furthermore, our findings suggested that the acetylation site at position 97 plays a crucial role in regulating the ubiquitination process, acting as a switch to turn it on or off.

We have updated our results and statements.

5) The specificity of the KAc antibody used needs to be better defined.

We regret for not included the antibody information in the prior version of the manuscript. The pan Kac antibody is a specific Kac antibody that has been widely employed in numerous acetylation studies (Please refer to the following two recent articles that have utilized the pan antibody:). The antibody information has been updated in the modified manuscript. Please see line 837-838.

1. Li, et al. Fungal acetylome comparative analysis identifies an essential role of acetylation in human fungal pathogen virulence[J].Communications Biology.2019

2. Liu, et al. SIRT7 couples light-driven body temperature cues to hepatic circadian phase coherence and gluconeogenesis[J].Nature Metabolism.2019

6) The authors should calculate the MIC using standard methods.

We highly value these crucial suggestions. In the revised manuscript, we have conducted the drug inhibitory experiments again, employing the previously established technique. Additional quantitative results have been provided. Please see Figures 1b, 2e, 8e-8i, and line 562-580.

The revised manuscript will be sent back to the reviewers for re-review so please make sure you answer all the points above and below if you resubmit to eLife.Reviewer #1 (Recommendations for the authors):1. It would greatly enhance the paper if the authors could show that the isw1 mutant is resistant to antifungal treatment in mouse models. Given that the mutant is fully virulent, the difference in survival or fungal burden after drug treatment would indicate the potential role of Isw1 during antifungal therapy. That would enhance the impact of the research showing the potential clinical relevance of Isw1 regulation.

We express our gratitude for your significant and insightful feedback. We concur with the reviewer's assertion that understanding the control of Isw1 in the context of systemic infection is of utmost importance in terms of its potential clinical significance. In the revised manuscript, a series of experiments were conducted to validate the *in vivo* drug resistance of Isw1 regulation. Initially, we conducted experiments to ascertain that Isw1 does not function as a virulence regulator. This was achieved by evaluating the levels of capsule formation, melanin synthesis, and cell proliferation in response to adverse environmental conditions. The data presented in this study indicate that the protein levels of Isw1 are not subject to regulation by environmental pressures. Additionally, no growth abnormalities were observed in the isw1 mutant strain under these conditions. Next, we conducted an experiment to assess the involvement of the isw1 mutant strain in fungal colonization in mice. The findings revealed that the fungal load of the isw1 mutant strain was similar to that of the wildtype strain. Hence, the isw1 mutant exhibits suitability for subsequent analysis. Subsequently, mice were subjected to infection with either wildtype or mutant fungal cells, followed by the administration of chemotherapeutic agents. The data presented in our study revealed a significant increase in fungal load in the isw1 mutant cells as compared to the wildtype cells following the administration of 5-FC or FLC in mice. Hence, the updated results provide substantial evidence to support the notion that Isw1 plays a pivotal role in the regulation of drug resistance, both *in vitro* and *in vivo* organisms. Please see Figure 3 and line 234-250.

2. The MICs shown in Figure 1b and Figure 6f were not done according to standard protocols. Although the spotting assays are nice because they are visual, the authors need to use the accepted standard protocol to determine MICs of antifungals for both the mutants and the clinical isolates.

We appreciate this essential comments. In the revised manuscript, we have redone all the drug inhibitory tests using protocol described previously. More quantitative results have been updated. Please see Figures 1b, 2e, 8e-8i, and line 562-580.

3. It's surprising that all the clinical isolates used in this study showed altered Isw1 levels. The authors could include clinical isolates with known mechanisms of resistance for comparison.

We express our genuine gratitude for this comment. Firstly, the development of drug resistance in fungi is a complex mechanism that encompasses various regulators, including mutations in crucial gene sets and regulatory pathways. In addition, fungal cells employ distinct mechanisms of resistance to effectively counteract various categories of antifungal drugs. For instance, fungal cells engage different processes to develop resistance to 5-FC and FLC. Hence, the inclusion of one or more resistance players may significantly introduce additional variables, thereby impeding the study. Moreover, considering the extensive repertoire of known resistance mechanisms in fungi, the analysis of Isw1's function would be further complicated. Furthermore, this research, conducted in collaboration with local hospitals, has successfully found multiple strains of *C. neoformans* that exhibit resistance (as shown in Supplementary File 1 sheet2). However, it is worth noting that these strains do not possess any recognized mechanisms of resistance. In other words, the sequencing analysis conducted on these strains did not identify any mutations in genes known to be associated with drug resistance (see to Supplementary File 5 for more details). Although our analysis did not include any clinical strains with known mechanisms of resistance, it has effectively illustrated a significant association between Isw1 and drug resistance in the laboratory standard strain, H99. Moreover, through the manipulation of Isw1 protein levels in the clinical strains, we have demonstrated in great detail that the concentration of Isw1 protein has the ability to impact the susceptibility of cells to antifungal drugs. In response to this worry, our group has undertaken the initiative of expanding our clinical collections over a period of six months. The regulation of Isw1 was examined in a total of twelve clinical isolates. In this study, Isw1-Flag constructs were produced in an additional six clinical isolates, namely CDLC4, CDLC6, CDLC37, CDC43, CDLC100, and CDLC141. The protein expression data demonstrated the presence of three separate categories. 1. Strains exhibiting traits of multidrug resistance demonstrated significantly reduced levels of Isw1-Flag. The strains encompassed in this set are CDLC15, CDLC25, CDLC61, CDLC62, and CDLC98. 2. On the other hand, strains that demonstrated susceptibility to drugs revealed substantial protein expression levels of Isw1-Flag. Prominent instances of these strains include CDCL120, CDCLC6, CDCL37, CDCLC43, and CDLC100. Two clinical isolates that displayed resistance to antifungal medications, yet had high quantities of the Isw1 protein. Subsequent examination of these strains revealed that the levels of Isw1 exert regulatory influence over drug resistance in those strains. In the context of the CDCLC141 study, it is seen that despite the presence of a higher level of Isw1, the expression of a stable form of Isw1 actually diminishes drug resistance. Hence, the Isw1 regulatory axis might be considered a prototypical clinical phenomena. Please see figure 8, supplementary file 5 and line 373–416.

4. The authors believe that reduction in 5-FC uptake contributes to the resistance of the isw1 mutant to this drug. Given that isw1 mutation altered the expression of many genes (but none of the changes of the transporter genes are huge as the log2 FPKM is within 1.5, Figure 1C) and that there is only one known transporter Fcy2 involved in the intake of 5-FC, this presents an ideal case for the authors to demonstrate that reduced uptake is indeed the underlying resistance mechanism.

We much value the insightful feedback provided by the reviewer. However, it is our contention that this expert may have misconstrued our intracellular quantification results for FLC. In both the prior and revised manuscript, we included a quantitative analysis of intracellular fluconazole concentration, but not of 5-FC. Initially, we observed an upregulation in the expression levels of genes responsible for encoding drug pumps. Specifically, we identified a total of 12 pumps, consisting of 7 members from the ABC family, 2 efflux pumps, and 3 pumps from the MSF family. In order to conduct a more accurate assessment of the control of Isw1, it is advisable to create a strain whereby the genes responsible for the functioning of these 12 pumps are disrupted, within the background of an isw1 deletion strain. Nevertheless, due to the extensive number of genes implicated and the potential variations in pump combinations that may contribute to the resistance mechanism, the suggested method is deemed unsuitable for procurement. Alternatively, we inquired about the potential involvement of Isw1 in the modulation of drug efflux. In this regard, it may be feasible to assess the intracellular concentration of FLC as a means to more accurately ascertain the drug tolerance state of the isw1 mutant strain. No recognized receptors for fluconazole (FLC) have been found in fungal organisms, and drug efflux pumps have been directly associated with the development of drug tolerance. Hence, from the quantification of intracellular FLC concentration and the investigation of pump gene expression, it is highly probable that Isw1 plays a role in regulating drug resistance by modifying drug efflux. In both the prior and current iterations of the work, we have not conducted quantitative analysis of intracellular 5-FC concentration. This is due to the fact that 5-FC functions as a prodrug, undergoing rapid metabolism and conversion into several downstream molecules. Consequently, the quantification of 5-FC becomes more challenging. However, a substantial body of prior literature has comprehensively illustrated the association between 5-FC and resistance genes, including Fcy2, Fcy1, and other significant genes. Significantly, Isw1 not only governs the entry of drugs (Fcy2), but it also modulates both canonical and non-canonical mechanisms of 5-FC drug resistance pathway, encompassing four pivotal genes. Consequently, we have utilized a comprehensive integration of biochemical, genetic, and molecular biological experimental evidence, along with corroborating published findings, to demonstrate that Isw1 exerts control over FLC resistance by regulating the expression of drug pumps, and governs 5-FC resistance by modulating the expression of 5-FC metabolic pathways. The manuscript has been revised. Please see line 145-153 and 167-178.

Reviewer #2 (Recommendations for the authors):General comments:The term protein posttranslational modification (PPTM) is more commonly referred to as post-translational modification (PTM). The authors use PPTM throughout this manuscript. The authors could consider changing "PPTM" to "PTM".

All PPTMs have been changed to PTMs.

The authors' use of "increase", "induce", or similar wording is confusing in some instances. For example, in Lines 197-198 the authors suggest that deacetylase inhibitors "increase acetylation levels" (see also Lines 235, 283, and 284 for additional examples). The authors should consider rewording these examples and reviewing the manuscript for similar examples to clarify their results.

The manuscript has been appropriately updated in accordance with the given instructions.

The authors find that deletion of ITC1, which is known to interact with Isw1 homologs in other systems, results in increased antifungal resistance. However, loss of Itc1 does not fully phenocopy loss of Isw1 (see Figure 1—figure supplement 2 panel B). This is likely due to the activity of other Isw1 interacting proteins. The authors are encouraged to review recent literature on other organisms (particularly other fungi, such as *Neurospora crassa*) on other potential interacting proteins.

We acknowledge the significance of the conservation of Isw1 across many species. The regulatory pathway of Isw1 was initially discovered in the model organism *Saccharomyces cerevisiae*, which possesses two paralogs, namely Isw1 and Isw2, as a result of a whole genome duplication event(Kellis M, 2004; Tsukiyama T, 1999; Wolfe KH, 1997). Due to the absence of a complete genome duplication event in *C. neoformans*, its genome solely contains a single copy of the ISW gene. Prior research conducted on *Saccharomyces cerevisiae* has provided evidence that the ISWI complex is composed of several subunits, namely Isw1, Ioc proteins, Itc1, Chd1, and Sua7 (Mellor J, 2004; Smolle M, 2012; Sugiyama and Nikawa, 2001; Vary JC Jr, 2003; Yadon AN, 2013). The genome of *C. neoformans* was examined, and it was observed that the *IOC* gene family could not be identified. This suggests that the *IOC* gene family has likely undergone an evolutionary loss in *C. neoformans*, as indicated on the FungiDB website. The genome of *C. neoformans* harbors three distinct genes, namely Itc1, Chd1, and Sua7. In order to comprehensively investigate the cryptoccocal ISWI complex, we conducted a methodical Isw1-Flag protein immunoprecipitation procedure, which was subsequently followed by Mass-Spec analysis. In the present study, a total of 22 proteins that interacted with Isw1 were discovered. Among these proteins, 11 have been previously reported to be associated with the regulatory networks including Isw1. In the mass spectrometry results, Itc1 was found to be co-immunoprecipitated with Isw1. Although the Mass-Spec analysis did not reveal the presence of Chd1 and Sua7, our study demonstrated that Chd1 can be coimmunoprecipitated with Isw1 through co-IP and immunoblotting techniques. However, no interaction between Isw1 and Sua7 was established utilizing any of these methods. In order to gain a deeper understanding of the involvement of Chd1 and Itc1 in the regulation mechanism of Isw1 in multidrug resistance, we created strains with disrupted *chd1Δ* and *itc1Δ* genes. The only strain that exhibited a drug resistance phenotype similar to that of the *isw1Δ* strain was the *itc1Δ* strain. The data presented in this study indicate that there has been evolutionary divergence in the Isw1-protein complexes between the species *C. neoformans* and *S. cerevisiae*. The manuscript has undergone significant modifications. Please see Figure 2 and text line 208-232.

The authors found that deletion of CDC4 in the Isw1 K97R background results in reduced resistance to antifungals. It also appears that the deletion of CDC4 in the Isw1 K97R background impacts general growth as shown on YPD in Figure 5 (panel A). Did the authors examine the deletion of CDC4 in the wild-type background? How does deletion of CDC4 in wild type impact general growth and resistance to antifungals?

We express our gratitude for this comment. In the revised publication, we created a *cdc4Δ* deletion strain and subsequently shown that the absence of *cdc4Δ* does not exert any discernible impact on drug resistance. It is postulated that the observed phenomenon may be attributed to the presence of other E3 ligases that modulate the stability of the Isw1 protein. This speculation arises from the observation that, even in the absence of Cdc4 in the K97R mutant strain, a certain degree of drug resistance growth is still exhibited. The manuscript has been revised to incorporate this information. Please see Figure 7—figure supplement 1b, and line 356.

The authors state that "results showed an interaction between Cdc4 and Isw1 K97R butnot between Cdc4 and Isw1 K97Q" (Line 269) in reference to data shown in Figure 5 (panel E). However, it appears that Cdc4 and Isw1 K97Q show a very weak interaction based on the co-IP data. The authors should address this discrepancy.

We concur with the assertion that the K97Q mutation resulted in a reduction in the strength of the interaction between Isw1 and Cdc4, rather than a complete blockade. The manuscript has been revised in accordance with the suggested revisions. Please see line 362-367.

The authors are encouraged to thoroughly review and revise the "Material and Methods" section and figure legends of the manuscript to ensure methods are described in sufficient detail to repeat, citations are provided when needed, and figures are adequately described. For example: (1) although the manuscript relies heavily on the quantification of western blot data the authors do not describe how this quantification was performed, and (2) the authors do not describe how RNA-seq data was processed or mapped (see Line 559) or provide a citation for DESeq2 package (see Line 562).

We have included all necessary details and references in the Material and Methods. Please see line 820-824 and line 858. (Gao X, 2022; Love MI, 2014).).

It is not clear the number of biological replicates used or if experiments were repeated for some experiments. For example: (1) in Figure 3 panels A and B and related source data, it appears these experiments were run once with one biological replicate, and (2) in Figure 3 panel F and related source data, it appears that western blots for two replicates are shown, although the data for three replicates are plotted. Could the authors clarify the number of replicates used and if experiments were repeated?

We apologize for not providing the necessary replicate information. We have modified the text, and including the raw data for biological replicates in the source data. Please see line 1360 and source data figure 5 source data 8.

Specific comments:Line 68: The authors should rewrite/edit Line 68 for clarity.

Please see line 51-53.

Line 69: The citation, "(2017)", is incomplete. The authors should correct the citation here and check throughout the manuscript for additional instances of incomplete citations (e.g., Line 74).

Please see line 53, 60.

Line 103: It appears the Florent et al., 2009 citation is incorrectly used here, as the referenced study investigated antifungal resistance in Candida lusitaniae.

Please see line 89

Line 134: The authors could consider briefly describing why they chose to study Isw1 in Cryptococcus.

Please see line 128–134.

Line 136: The authors should specify that the isw1 mutant was complemented with an ISW1 allele tagged with FLAG.

Please see line 136.

Line 154: The authors state "To test if this increase affects drug uptake,…"; however, the experiment described doesn't distinguish between decreased drug import and increased drug export. This should be rewritten for clarity.

Please see line 162–166.

Line 180: The authors should rewrite/edit Line 180 for clarity.

Please see line 260-262.

Line 191: The authors should rewrite/edit Line 191 for clarity.

Please see line 273.

Line 200: The authors should rewrite/edit Line 200 for clarity and define the abbreviation for DNA binding domain as only "DBD" is used in Figure 2 (panel F) to avoid confusion. Additionally, the authors should describe why certain acetylated residues were chosen and others were not.

Please see Figure 1i and line 197–201.

Lines 201-202: The authors should revise the statement "to mimic a fully acetylated Isw1" for clarity and correctness, as there appear to be additional acetylated lysine residues in Isw1.

Please see line 282-284.

Lines 208-209: The authors should rewrite/edit Lines 208-209 for clarity.

Please see line 291-293.

Lines 220-221: The authors should rewrite/edit Lines 220- a221 for clarity.

Please see line 305–306.

Line 246: Data presented in this study suggest 15 residues of Isw1 are ubiquitinated but only six were mutated. The authors should provide reasoning for selecting certain ubiquitination sites in the manuscript. Specifically, it is curious that the authors did not test K98 as the proximity to K97 could affect acetylation or vice versa. Can the authors explain why K98 was not tested? Additionally, the authors state that "five failed to affect drug-resistant growth phenotypes". However, it seems that two of the six tested residues had an impact on drug resistance in the K97R background. The authors should address this discrepancy.

In light of the above criticism, we have undertaken further measures to induce a mutation in the K98 site, resulting in the development of the R variant within the genetic background of the K97 mutation strain. The findings of our study indicate that modifications to the K98 sites do not have any discernible effect on drug resistance. In the present study, our research aimed to investigate the interplay between acetylation and ubiquitination in the modulation of drug resistance. In order to mitigate the substantial workload associated with simultaneously altering all 15 ubiquitination sites, our research team made the decision to concentrate our efforts on the sites neighboring the K97 site. This strategic approach was based on the logical assumption that these adjacent sites possess the greatest potential for being influenced by the K97 sites. There is a possibility that additional ubiquitination sites may play a significant role in regulating the stability of the Isw1 protein. The study's findings indicate the presence of reciprocal control between acetylation and ubiquitination. The current data obtained in our research substantiates this principle and offers compelling evidence in support of this conclusion. The manuscript has been revised to provide a description of the process involved in selecting Ub locations. Please see line 333-340.

Lines 250-252: The authors should rewrite/edit Lines 250-252 for clarity.

Please see line 341-344.

Line 277: "majority was" should be changed to "majority were" in this instance.

Please see line 395–404.

Lines 307-308: Gcn5 is not a deacetylase but is the catalytic subunit of the histone acetyltransferase complexes, ADA, and SAGA. The authors should address this discrepancy.

Please see line 434-436.

Line 313: The authors use "the acetylation-Isw1-ubiquitination axis" here and use the "Isw1 acetylation-ubiquitin-proteasome regulation axis" above (see Line 289). The authors could consider using consistent wording throughout the manuscript for clarity.

We have modified text. Please see line 440.

Line 315: It appears the authors use a lowercase "L" instead of an uppercase "I" for Ioc3 and Ioc4. The authors should address this discrepancy.

We have modified text. Please see line 442.

Lines 320-323: The authors do not directly demonstrate that transporter-encoding genes upregulated in the ISW1 deletion strain are responsible for reduced FLC in cells or if the reduction of FLC in cells is caused by decreased drug import and/or increased drug export (see comment for Line 154 above). Relatedly, it should be noted that Figure 1 (panel C) indicates multiple transporter-encoding genes are downregulated in the ISW1 deletion strain, and it is not clear if the authors describe and/or discuss this in the text. With these points in mind, the authors should rewrite Lines 320-323 for clarity.

We have modified the text. Please see line 455–459.

Lines 324-327: The authors should rewrite/edit Lines 324-327 for correctness and clarity. For example, the authors state "Isw1 is a transcription activator for genes responsible for resistance to FLC"; however, it seems that genes involved in FLC resistance are upregulated in the absence of Isw1, which means Isw1 activity represses expression of such genes. The authors are also encouraged to keep in mind that Isw1 is an ATP-dependent chromatin remodeling factor and not a transcription factor.

We have modified this. Please see line 464-468.

Figure 2 panels B and D: The authors should define CHX in the figure legend, i.e., "cycloheximide (CHX)".

We have modified this. Please see line 1258 and 1264.

Figure 4 panel E: It appears the images showing growth on 16 ug/ml FLC and 20 ug/ml FLC may be mislabeled. The authors should review and correct if needed.

We have modified this. Please see Figure 6e.

Reviewer #3 (Recommendations for the authors):A) How did the author identify ISWI as a candidate gene important for regulating drug resistance in Cryptococcus?

We express our gratitude for this criticism. The identification of Isw1 was conducted as a subsequent study to our previously published data (Li Y, 2019). In prior research, we conducted a comprehensive analysis of the acetylome in *C. neoformans*, and then developed knockout strains to investigate the relationship between acetylation and fungal pathogenicity. The Isw1 mutant has been discovered as a modifier of drug resistance. Please see line 129-134.

B) The budding yeast, *S. cerevisiae*, expresses two homologs of ISWI, Isw1p, and Isw2p. Isw2p associates with Itc1p (Sugiyama and Nikawa, 2001) The Isw1p forms two distinct complexes: one subcomplex is formed by Isw1p, Ioc2p, and Ioc4p and a second by Isw1p and Ioc3 (Vary Mol Cell Biol 2003). Does Cryptococcus contain two ISWI genes? If so, why did the authors focus just on the Isw(2?)-Itc1p proteins? Does Cryptococcus encode for Ioc2, Ioc3, and Ioc4? The authors should clarify this point.

We acknowledge the significant concern raised over the conservation of Isw1 across several species. The regulatory mechanism of Isw1 was initially discovered in the model organism *Saccharomyces cerevisiae*. This process involves two paralogs, Isw1 and Isw2, which emerged as a result of the complete genome duplication event (Kellis M, 2004; Tsukiyama T, 1999; Wolfe KH, 1997). Because *C. neoformans* has not gone through the complete genome duplication event, its genome only encodes one copy of ISW gene. Prior research conducted on *Saccharomyces cerevisiae* has provided evidence that the ISWI complex is comprised of several subunits, namely Isw1, Ioc genes, Itc1, Chd1, and Sua7 (Mellor J, 2004; Smolle M, 2012; Sugiyama and Nikawa, 2001; Vary JC Jr, 2003; Yadon AN, 2013). Upon doing a thorough examination of the genome of *C. neoformans*, our investigation yielded negative results in terms of identifying the *IOC* gene family. This absence likely suggests an evolutionary loss of the *IOC* gene family in C. neoformans, as suggested on the FungalDB website. The genome of *C. neoformans* has the genes Itc1, Chd1, and Sua7. While we concur with the aforementioned statement on the capability of Mass-Spec data to elucidate potential protein-protein interactions and aid in the identification of subunits within the ISWI complex, it is important to acknowledge that the PTM Mass-Spec methodology is solely employed for the purpose of identifying potential sites of protein modification. In order to comprehensively investigate the cryptoccocal ISWI complex, we conducted a standardized Isw1-Flag protein immunoprecipitation procedure, followed by Mass-Spec analysis. In the present study, a total of 22 proteins that interact with Isw1 were found in our experimental data. Among these proteins, 11 have been previously reported to be associated with the regulatory networks including Isw1. In the mass spectrometry results, the protein Itc1 was found to be co-immunoprecipitated with the protein Isw1. Although the Mass-Spec analysis did not reveal the presence of Chd1 and Sua7, our study demonstrated that Chd1 can be coimmunoprecipitated with Isw1 through the utilization of co-IP and immunoblotting techniques. However, no interaction between Isw1 and Sua7 was shown utilizing any of these methods. The data presented in this study indicate that there has been evolutionary divergence in the Isw1-protein complexes between *C. neoformans* and *S. cerevisiae*. The manuscript has undergone significant modifications. Please see Figure 2 and line 206-232.

C) The itc1 deletion strain is barely resistant to FLC (Figure S2): MIC for isw1 δ and itc1 δ strains should be calculated.

We highly value these crucial remarks. In the revised manuscript, we have conducted the drug inhibitory tests once again, following the previously demonstrated technique (Xie J et al., 2012). Additional quantitative results have been provided. Please refer to Figures 2e and Line 224-228. The strain with the deletion of itc1 gene exhibits resistance to antifungal drugs.

D) Did the authors perform transcriptomic analysis of WT and isw1 deleted cells without FLC? The material and methods section suggests that this experiment has been performed (Pg 18 line 544). However, these RNA-seq data are not shown in Figure 1. This is a critical experiment as it establishes whether ISWI is an activator or a repressor of gene expression.

In the revised manuscript, we conducted a transcriptome analysis of both wildtype and isw1 deletion strains in the absence of FLC treatment. When comparing the data collected under two different settings, namely with and without FLC treatment, we observed a distinct difference in the regulation of gene expression between these two conditions. In the case of the isw1 deletion strain subjected to FLC treatment, a total of 21 genes, encompassing the ABC/MFS family and efflux pumps, exhibited substantial alterations in gene expression. Specifically, 9 genes were downregulated while 12 genes were upregulated. Conversely, when FLC supplementation was not provided, only 9 genes demonstrated changes in gene expression, with 3 genes being downregulated and 6 genes being upregulated. Hence, the Isw1 protein is essential for the activation of specific genes, while concurrently exerting a repressive role on other genes. The manuscript has been revised to accurately and comprehensively depict our findings. Please see Figure 1c, Supplementary File 2 and line 145-153.

E) Isw1-FLAG pulldown- It is strange that the authors dive into Isw1 post-translational modifications without describing the pull-down and characterising the protein complexes. What is the efficiency of the pull-down? This could be shown by Silver staining+ Western blot. What are the Iswi1-interacting proteins? Are the ISWI-protein complexes conserved in Cryptococcus?

We express our gratitude for this comment. Pull-down studies are frequently employed as a prevalent method for analyzing post-translational modifications (PTMs) of a particular protein in cases where antibodies specific to PTM sites are not accessible. The used methodology involves the utilization of protein pull-down technique in conjunction with pan antibody detection for the purpose of acetylation analysis. This approach aims to enhance the concentration of the target protein by employing commercially available beads specifically designed for epitope tag isolation. In our investigation, Flag beads (Σ) were utilized for this purpose. Subsequently, specialized antibodies, such as a pan anti-lysine acetylation antibody, are employed to ascertain the overall amounts of acetylation in Isw1. In order to investigate the protein complexes associated with Isw1 in the organism *C. neoformans*, our study employed a series of scientific methodologies including bioinformatic analysis, protein co-immunoprecipitation followed by Mass Spectrometry, immunoblotting experiments, and gene disruption analyses. Prior research conducted on *Saccharomyces cerevisiae* has identified several proteins that interact with Isw1, namely Ioc proteins, Itc1, Sua7, and Chd1(Mellor J, 2004; Smolle M, 2012; Sugiyama and Nikawa, 2001; Vary JC Jr, 2003; Yadon AN, 2013).. However, it has been observed that the regulatory network of the cryptococcal Isw1-protein associated complex exhibits substantial divergence when compared to that of *S. cerevisiae*. The absence of the *IOC* gene family in the genome of *C. neoformans* indicates a distinct regulatory pattern of Isw1 between the two species. Furthermore, the detection of protein interaction between Isw1 and Sua7 in *S. cerevisiae* has not been seen through the utilization of Mass-Spec and Co-IP assays in *C. neoformans*. The connections between Isw1-Itc1 and -Chd1 were indeed confirmed in *C. neoformans*, indicating a certain degree of preservation of the ISWI complex machinery throughout fungal evolution. Subsequently, we have undertaken additional efforts to elucidate the role of Chd1 and Itc1 in the regulation of the multidrug resistance mechanism. It is noteworthy that among the tested strains, only Itc1 exhibited a reciprocal regulation of drug resistance in *C. neoformans*. This particular phenotype bears resemblance to that seen in the cryptoccocal *isw1Δ* strain. In summary, the ISWI regulatory mechanism of cryptococcal organisms exhibits a distant relationship to that found in *S. cerevisiae*. Please see Figure 2 and line 206-232.

F) For all the FLAG-Western Blots: No FLAG control is missing.

In this investigation, we utilized the Flag antibody produced by Σ, a well-established manufacturer that has been widely exploited in previous research, leading to a substantial amount of publications. The Σ antibody does not generate any non-specific bands when tested against both the wildtype strain (H99) and the FLAG strains. In order to provide further clarification regarding the specific antibody, our study incorporates a series of studies comprising eight biological duplicates and three sets of complete blots obtained from three different exposure durations. The presence of Isw1 was observed at around 130 kDa, while no discernible band corresponding to the H99 strain was found. Hence, this antibody yields outcomes that are both specific and dependable.

**Author response image 1. sa2fig1:** 

G) For all the FLAG Pulldowns: Input is missing.

In the manuscript, all inputs were provided as depicted in Figure 5a and 5b. This comment suggests that the observed acetylation results are obtained without any inputs. The regular input samples are utilized to investigate the presence of proteins of interest. In the context of PTM analysis, the detection method involves the utilization of a pan anti-acetyllysine antibody for the purpose of PTM detection. Routine inputs can only generate a broad representation of the overall acetyllysine levels throughout the cell. However, these input samples cannot serve as a control group for accurately quantifying specific target proteins or post-translational modifications (PTMs). Please refer to the figure provided below, which displays the prior publications as well as the input samples utilized for the detection of acetylation.

H) What does the anti-Kac detect? Iswi1 acetylation? Did the authors raise the antibody? Histone acetylation? I cannot find this info in the Material and Methods. Assuming that the antibody detects Isw1 acetylation, I am not convinced that "the presence of antifungal agents strongly repressed acetylation levels" (Line 196) as reduced protein acetylation levels are observed only when FLC treatment is combined with TSA/NAM treatment.

We regret the omission of the antibody information in the prior iteration of the manuscript. The pan Kac antibody is a highly specific antibody targeting lysine acetylation (Kac), which has been extensively employed in numerous studies involving acetylation analysis. Please refer to the provided references for further information. The revised manuscript includes updated information regarding the antibodies. The statement has been updated. Please see line 836-838.

1. Qiutao Xu, et al. ROS-stimulated Protein Lysine Acetylation Is Required for Crown Root Development in Rice[J].Journal of Advanced Research.2022.

2. Nan Liu, et al. HDAC inhibitors improve CRISPR/Cas9 mediated prime editing and base editing[J].Molecular Therapy-Nucleic Acids.2022.

3. Yi Fang, et al. Histone crotonylation promotes mesoendodermal commitment of human embryonic stem cells[J].Cell Stem Cell.2021.

4. Li, et al. Fungal acetylome comparative analysis identifies an essential role of acetylation in human fungal pathogen virulence[J].Communications Biology.2019.

5. Shun Tu, et al. YcgC represents a new protein deacetylase family in prokaryotes[J].*eLife*.2015.

**Author response image 3. sa2fig3:** 

I) Although the mutation analysis demonstrates that K97 plays a role in the drug response- data demonstrating that acetylation of this residue regulates this process are missing. Could the mutation affect ISWI folding? The crystal structure of several ISWI proteins has been solved and it should be relatively easy to predict the effect of K97 mutations on ISWI protein folding.

In light of the aforementioned comment, a comprehensive structural algorithm analysis was conducted on both Isw1 wildtype and mutant variants. The results of the prediction analysis did not reveal any substantial changes in the folding of Isw1.

J) Protein poly-ubiquitination (normally linked to proteosome-degradation) leads to higher molecular weight protein species in a Western blot analysis. Therefore, it is puzzling that these high molecular weight species are not detected, especially after MG132 treatment.

We express our gratitude for this comment. In theory, the process of poly-ubiquitination results in the formation of high molecular-weight proteins that exhibit smear patterns. However, when employing the immunoblotting technique to analyze poly-ubiquitination proteins, it is common to observe the detection of the apo form of the target proteins (i.e., non-ubiquitinated forms of the protein). This is mostly due to the low abundance of ubiquitinated protein species. It is also crucial to note that the presence of a smear in a detection assay does not provide definitive evidence of ubiquitination, as the smear may be attributed to various other factors. Hence, the identification of ubiquitinated proteins necessitates an additional enrichment step, such as protein immunoprecipitation assay, followed by the detection of ubiquitin through the utilization of anti-ubiquitin antibodies. In addition, it should be noted that mass spectrometry is considered the most precise technique for the detection of ubiquitination. It is worth mentioning that the manuscript in its prior iteration included data obtained using MS analysis. In order to provide a more precise account of conventional techniques employed for the detection of ubiquitination using immunoblotting, we present below two illustrative instances of ubiquitination detection investigations that have been recently published. Both experimental protocols involved the utilization of the protein immunoprecipitation (IP) technique, followed by immunoblotting analysis employing anti-ubiquitin (UB) antibodies.

Study 1.Feeding induces cholesterol biosynthesis via the mTORC1–USP20–HMGCR axis. 2019, Nature, https://doi.org/10.1038/s41586-020-2928-y

Please see Figure 1 for HMGCR protein IP followed by Ub detection using Ub antibody

Study 2.USP10 strikes down b-catenin by dual-wielding deubiquitinase activity and phase separation potential, Cell Chemical Biology. https://doi.org/10.1016/j.chembiol.2023.07.016

Please see Figure E,F,H for Axin1 protein IP followed by Ub detection using Ub antibody

In our investigation, we have seen that the native form of Isw1 can be detected using immunoblotting, but the presence of high molecular weight protein species remains undetectable. This finding aligns with the outcomes reported in the aforementioned papers. In the revised manuscript, an attempt was made to investigate the impact of IP Isw1 and afterwards measure ubiquitination levels using commercially available mammalian Ub antibodies. However, these efforts did not yield successful results. In order to address the concerns raised in this statement, we successfully engineered a strain of C. neoformans capable of producing the human UBB1 protein. Experiments using ubiquitination were conducted. Three strains were subjected to testing, namely the wildtype strain, the ISW1K97Q strain, and the ISW1K97R strain. The cells were subjected to incubation in medium that were either supplemented with MG132 or lacking MG132. The experimental protocol involved conducting protein pull down experiments utilizing Flag beads, followed by the subsequent identification and quantification of ubiquitin through the utilization of anti-human ubiquitin antibodies. Following an extended period of exposure, we successfully observed the presence of poly-ubiquitinated protein smear bands, which corresponded to the elevated molecular weight of poly-ubiquitinated Isw1. The results reveals that the presence of poly-ubiquitinated Isw1 species is observed exclusively in the ISW1K97R strain, while no such species are detected in the wildtype and ISW1K97Q strains. The presence of the K97Q mutant inhibits the ubiquitination of Isw1, so providing additional evidence to support the hypothesis that Isw1 is a protein that undergoes ubiquitination. Furthermore, our findings suggest that the acetylation site at position 97 plays a crucial role in regulating the ubiquitination process, acting as a switch to turn it on or off.

K) Is the Cdc4-Iswi interaction detected by Mass Spec Analysis?

In the revised manuscript, we conducted Isw1-Flag pull-down experiments followed by Mass spectrometry analysis. However, our findings did not reveal the presence of Cdc4. This can be attributed mostly to the limitations of the mass spectrometry technique, which resulted in insufficient coverage of the proteome. Despite recent advancements in proteomic technologies, the task of achieving a thorough coverage of an organism or sample's proteome remains tough. Certain proteins have the potential to evade detection or pose challenges in their identification, hence impeding a comprehensive comprehension of the entirety of the protein landscape. Nevertheless, our experimental approach including protein co-immunoprecipitation (co-IP) and subsequent immunoblotting provided evidence of the interaction between Cdc4 and Isw1. Please see Table S4.

L) Is ISWI function in drug resistance linked to its chromatin remodelling activity?

In order to investigate the potential role of Isw1 chromatin activity in the modulation of multidrug resistance, we conducted protein truncation experiments. Specifically, we genetically deleted the DNA binding domain, helicase domain, and SNF2 domain, which have been previously shown to regulate Isw1 chromatin activity in the model organism *S. cerevisiae* (Grune T, 2003; Mellor J, 2004; Pinskaya M, 2009; Rowbotham SP, 2011). The data demonstrates that all shortened variants of Isw1 mutants had a growth phenotype characterized by multidrug resistance, which is consistent with the growth phenotype observed in the deletion strain *isw1Δ*. Furthermore, the levels of gene expression observed in these strains were found to be similar to those observed in the deletion strain *isw1Δ*. This finding provides evidence that the regulation of the drug resistance mechanism is actually influenced by these critical domains involved in modifying chromatin activity. In addition, the Isw1-Flag strain was utilized to conduct chromatin immunoprecipitation and PCR experiments, which revealed that Isw-1 exhibits the ability to directly bind to the promoter regions of target genes. The findings from this collective analysis of the revised data provide substantial evidence supporting the notion that the Isw1 chromatin activity plays a crucial role in modulating its protein function, acting as a central regulator of drug resistance in *C. neoformans*. Please see Figure 1g, 1h, 1i and line 186-199.

M) Does ISWI interact with chromatin? If so, which are ISWI-target genes? Does drug treatment modulate chromatin binding?

To effectively tackle this concern, we have pursued two distinct approaches to demonstrate the chromatin regulatory effects of Isw1. In this study, the DNA binding domain was deliberately removed through genetic manipulation. The data presented indicates that the Isw1 mutants with shorter variations exhibited a growth phenotype that was characterized by multidrug resistance. This growth phenotype correlates with the growth phenotype obtained in the *isw1Δ* deletion strain. Additionally, it was observed that the levels of gene expression in the strain were comparable to those detected in the deletion strain *isw1Δ*. This discovery offers empirical support for the notion that the control of the drug resistance mechanism is indeed impacted by the DNA binding capability of Isw1. Furthermore, the Isw1-Flag strain was employed to perform chromatin immunoprecipitation and PCR assays, demonstrating the direct binding capacity of Isw1 to the promoter regions of target genes. The results obtained from this comprehensive analysis of the revised data offer significant evidence for the proposition that Isw1 interacts with chromatin and that its chromatin activity plays a pivotal role in modulating its protein function. This interaction serves as a central regulatory mechanism for drug resistance in *C. neoformans*. Furthermore, a transcriptome analysis was performed on both wildtype and isw1 deletion strains in the absence of FLC therapy. Upon comparing the results obtained from two unique experimental settings, specifically those with and without FLC administration, a notable disparity in the control of gene expression between these two situations was identified. In the context of the isw1 deletion strain exposed to FLC treatment, a set of 21 genes, including those belonging to the ABC/MFS family and efflux pumps, displayed significant changes in their gene expression patterns. In particular, a total of 9 genes exhibited downregulation, whilst 12 genes displayed upregulation. In contrast, in the absence of FLC supplementation, a total of 9 genes exhibited alterations in gene expression, with 3 genes showing downregulation and 6 genes showing upregulation. Therefore, the Isw1 protein plays a crucial role in the activation of certain genes, while simultaneously having a suppressive effect on other genes. Hence, the Isw1 undergoes a reconfiguration of its regulatory apparatus in response to drugs. Despite that the performance of ChIP-seq analysis was necessary in this study, it was observed that the treatment of fungal cells resulted in a notable decrease in the abundance of the Isw1 protein. This decrease can be attributed to the activation of Isw1 protein degradation. Consequently, there was an insufficient amount of Isw1 protein available for successful enrichment and subsequent ChIP-seq analysis (please see Figure 4a and 4c). However, the data collected collectively have demonstrated the idea that Isw1 serves as a crucial master regulator of drug resistance in *C. neoformans*. The text has undergone revisions in order to present our findings in a precise and thorough manner. Please see Figure 1c, 1g, 4a, 4b, Supplementary File 2, and line 145-153, 186-192, 256-260.

Reference

Gao X, F.Y., Sun S, Gu T, Li Y, Sun T, Li H, Du W, Suo C, Li C, Gao Y, Meng Y, Ni Y, Yang S, Lan T, Sai S, Li J, Yu K, Wang P, Ding C. (2022). Cryptococcal Hsf3 controls intramitochondrial ROS homeostasis by regulating the respiratory process.. Nat Commun. *13*. doi:10.1038/s41467-022-33168-1.

Grune T, B.J., Eberharter A, Clapier CR, Corona DF, Becker PB, Muller CW. (2003). Crystal structure and functional analysis of a nucleosome recognition module of the remodeling factor ISWI. Mol. Cell *12*, 449-460. doi:10.1016/s1097-2765(03)00273-9.

Kellis M, B.B., Lander ES. (2004). Proof and evolutionary analysis of ancient genome duplication in the yeast *Saccharomyces cerevisiae*.. Nature. *428*, 617-624. doi:10.1038/nature02424..

Li Y, L.H., Sui M, Li M, Wang J, Meng Y, Sun T, Liang Q, Suo C, Gao X, Li C, Li Z, Du W, Zhang B, Sai S, Zhang Z, Ye J, Wang H, Yue S, Li J, Zhong M, Chen C, Qi S, Lu L, Li D, Ding C. (2019). Fungal acetylome comparative analysis identifies an essential role of acetylation in human fungal pathogen virulence. Commun Biol *2*, 154. doi:10.1038/s42003-019-0419-1.

Love MI, H.W., Anders S. (2014). Moderated estimation of fold change and dispersion for RNA-seq data with DESeq2.. Genome Biol. *15*, 550. doi:10.1186/s13059-014-0550-8.

Mellor J, M.A. (2004). ISWI complexes in *Saccharomyces cerevisiae*. Biochim Biophys Acta., 100-112. doi:10.1016/j.bbaexp.2003.10.014..

Pinskaya M, N.A., Clynes D, Morillon A, Mellor J. (2009). Nucleosome remodeling and transcriptional repression are distinct functions of Isw1 in *Saccharomyces cerevisiae*.. Mol Cell Biol *29*. doi:10.1128/MCB.01050-08.

Rowbotham SP, B.L., Neves-Costa A, Santos F, Dean W, Hawkes N, Choudhary P, Will WR, Webster J, Oxley D, Green CM, Varga-Weisz P, Mermoud JE. (2011). Maintenance of silent chromatin through replication requires SWI/SNF-like chromatin remodeler SMARCAD1. Mol Cell. *42*, 285-296. doi:10.1016/j.molcel.2011.02.036.

Smolle M, V.S., Gogol MM, Li H, Zhang Y, Florens L, Washburn MP, Workman JL. (2012). Chromatin remodelers Isw1 and Chd1 maintain chromatin structure during transcription by preventing histone exchange.. Nat Struct Mol Biol. *19*, 884-892. doi:10.1038/nsmb.2312.

Sugiyama, M., and Nikawa, J. (2001). The *Saccharomyces cerevisiae* Isw2p-Itc1p complex represses INO1 expression and maintains cell morphology. J Bacteriol *183*, 4985-4993. doi:10.1128/JB.183.17.4985-4993.2001.

Tsukiyama T, P.J., Landel CC, Shiloach J, Wu C. (1999). Characterization of the imitation switch subfamily of ATP-dependent chromatin-remodeling factors in *Saccharomyces cerevisiae*. Genes Dev. *13*, 686-697. doi:10.1101/gad.13.6.686.

Vary JC Jr, G.V., Qin J, Landel CC, Kooperberg C, Bartholomew B, Tsukiyama T. (2003). Yeast Isw1p forms two separable complexes in vivo.. Mol Cell Biol. *23*, 80-91. doi:10.1128/MCB.23.1.80-91.2003.

Wolfe KH, S.D. (1997). Molecular evidence for an ancient duplication of the entire yeast genome. Nature *387*, 708-713. doi:10.1038/42711.

Yadon AN, S.B., Hampsey M, Tsukiyama T. (2013). DNA looping facilitates targeting of a chromatin remodeling enzyme.. Mol Cell. *50*, 93-103. doi:10.1016/j.molcel.2013.02.005..

[Editors’ note: what follows is the authors’ response to the second round of review.]

While the manuscript has been improved there are some remaining issues that need to be addressed, as outlined below:Reviewer #2 (Recommendations for the authors):General comments on revision: The revised manuscript is much improved overall. The authors have performed numerous additional experiments that strengthen the study. The majority of my initial comments have been addressed, and I thank the authors for carefully considering the comments and their responses. However, there are a few key points in the manuscript (largely associated with revisions or new data) that should be addressed to further strengthen the study and broaden its impact.

We express our sincere gratitude for your insightful feedback, particularly concerning the ISWI complex, which significantly contributes to the manuscript's improvement and facilitates readers' comprehension.

Prior reviewer comment: The authors demonstrate that Isw1 has a role in responding to antifungals in Cryptococcus. However, it is not clear if changes in Isw1 stability represent a general response to stress. This study would have benefited from experiments to test: (1) if levels of Isw1 change in response to other stressors (e.g., heat, osmotic, or oxidative stress) and (2) if loss of Isw1 impacts resistance to other stressors.Author response: A series of experiments were conducted to illustrate and measure phenotypic traits associated with virulence. These traits encompassed capsule formation, melanin synthesis, cell proliferation under stressful conditions, and Isw1 expression levels in response to diverse environmental stimuli. Please see Figure 3a, 3b, 3c, Figure 3—figure supplement 1 and line 237-241.Reviewer response: The authors provide new data to address the initial comment. The phenotypic data presented in Figure 3a, 3b, and 3c are convincing, although it should be noted that there appears to be a slight reduction in melanin production in Figure 3b. The data presented in Figure 3 —figure supplement 1 is less convincing. It is likely difficult to say "that Isw1 expression is not affected by external stress inducers" (line 237) considering data presented in Figure 1 —figure supplement 1 and Figure 4. Specifically, (1) ISW1 expression is not affected by antifungal treatment under the conditions tested and (2) changes in Isw1 levels associated with antifungal treatment are observed after inhibiting protein synthesis. It is thus plausible that changes in Isw1 levels under stress conditions could be missed in the assay presented in Figure 3 —figure supplement 1. The authors should change the text accordingly.

We highly appreciate your significant comment regarding the regulation of Isw1 protein levels in response to environmental conditions. In the previous revision, we performed protein quantification on Isw1-Flag under several stimuli and found that the total protein levels remained unchanged. Although we acknowledge that this experiment did not accurately represent the stability of the Isw1 protein under these specific conditions, the lack of any observed cell growth abnormalities in the spotting assays indicates that Isw1 is unlikely to play a significant role in controlling cellular responses to these stresses. As a result, we opted not to allocate additional resources towards conducting a more comprehensive investigation into the stability of Isw1 when protein synthesis is repressed. Additionally, we agree with the data about a slight decrease in melanin synthesis. The Results section has been modified to provide a more precise depiction of the findings. In addition, the Discussion section has been revised to point out the unique function of Isw1 in controlling fungal virulence factors and pathogenicity. It also emphasizes the significance of Isw1 in regulating drug resistance in both laboratory conditions and animal models. Please see lines 261-264, 267-269 and 496-507.

Prior reviewer comment: The authors find that deletion of ITC1, which is known to interact with Isw1 homologs in other systems, results in increased antifungal resistance. However, loss of Itc1 does not fully phenocopy loss of Isw1 (see Figure 1 —figure supplement 2 panel B). This is likely due to the activity of other Isw1 interacting proteins. The authors are encouraged to review recent literature on other organisms (particularly other fungi, such as *Neurospora crassa*) on other potential interacting proteins.Author response: We acknowledge the significance of the conservation of Isw1 across many species. The regulatory pathway of Isw1 was initially discovered in the model organism Saccharomyces cerevisiae, which possesses two paralogs, namely Isw1 and Isw2, as a result of a whole genome duplication event(Kellis M, 2004; Tsukiyama T, 1999; Wolfe KH, 1997). Due to the absence of a complete genome duplication event in C. neoformans, its genome solely contains a single copy of the ISW gene. Prior research conducted on Saccharomyces cerevisiae has provided evidence that the ISWI complex is composed of several subunits, namely Isw1, Ioc proteins, Itc1, Chd1, and Sua7 (Mellor J, 2004; Smolle M, 2012; Sugiyama and Nikawa, 2001; Vary JC Jr, 2003; Yadon AN, 2013). The genome of C. neoformans was examined, and it was observed that the IOC gene family could not be identified. This suggests that the IOC gene family has likely undergone an evolutionary loss in C. neoformans, as indicated on the FungiDB website. The genome of C. neoformans harbors three distinct genes, namely Itc1, Chd1, and Sua7. In order to comprehensively investigate the cryptoccocal ISWI complex, we conducted a methodical Isw1-Flag protein immunoprecipitation procedure, which was subsequently followed by Mass-Spec analysis. In the present study, a total of 22 proteins that interacted with Isw1 were discovered. Among these proteins, 11 have been previously reported to be associated with the regulatory networks including Isw1. In the mass spectrometry results, Itc1 was found to be co-immunoprecipitated with Isw1. Although the Mass-Spec analysis did not reveal the presence of Chd1 and Sua7, our study demonstrated that Chd1 can be coimmunoprecipitated with Isw1 through co-IP and immunoblotting techniques. However, no interaction between Isw1 and Sua7 was established utilizing any of these methods. In order to gain a deeper understanding of the involvement of Chd1 and Itc1 in the regulation mechanism of Isw1 in multidrug resistance, we created strains with disrupted chd1Δ and itc1Δ genes. The only strain that exhibited a drug resistance phenotype similar to that of the isw1Δ strain was the itc1Δ strain. The data presented in this study indicate that there has been evolutionary divergence in the Isw1-protein complexes between the species C. neoformans and *S. cerevisiae*. The manuscript has undergone significant modifications. Please see Figure 2 and text line 208-232.Reviewer response: The authors provide new data to data address the initial comment. This data greatly improves the manuscript. However, it also raises multiple concerns that must be addressed.1. The authors should be mindful that (1) there is not a single "ISWI Complex" in *S. cerevisiae* or other eukaryotes, but rather ISWI homologs form multiple complexes (e.g., Isw1a, Isw1b, and Isw2 in *S. cerevisiae*); and (2) the IOC (Iswi One Complex) genes encoding Ioc2, Ioc3, and Ioc4 are not a "gene family" per se as the proteins are quite different, and it may be difficult to readily identify divergent homologs in C. neoformans. Additionally, the authors should clarify what is meant by "canonical ISWI complex" (line 231).

We deeply appreciate your perceptive feedback, especially about the ISWI complex, as it greatly enhances the text and aids readers in understanding. The amended text now includes a comprehensive background explanation of the ISWI complex in several eukaryotes, including the suggested fungus species *Neurospora crassa*. This addition allows for a more thorough understanding of the ISWI complex homologs in fungi. After considering the Ioc comments, we conducted a thorough sequence analysis, examining both sequence homology and gene synteny. With the assistance of a reputable fungal genome database (http://fungidb.org), we discovered that the *C. neoformans* genome does not possess the ioc2, ioc3, and ioc4 genes. Please consult below for the screenshot from the fungal database, which depicts the Ioc2 as an exemplar and reveals that only 9 orthologs were detected in fungi. Same analyses were also performed for Ioc3 and Ioc4 (screenshots not shown). Thirty-five orthologs were found for Ioc3, which corresponds to Gene YFR013W according to fungidb.org. Additionally, twelve orthologs were found for Ioc4, corresponding to Gene YMR044W according to fungidb.org. Nevertheless, the ortholog lists for these three proteins did not include any transcripts from *C. neoformans*. Thus, it is probable that *C. neoformans* had gene loss in the *IOC* genes during its evolutionary process, while *S. cerevisiae* and other fungal species potentially acquired additional copies of *IOC* genes. We concur with the reviewer's assertion that studying the intricate ISWI holds significant value in elucidating the evolutionary processes and governing cellular reactions. Nevertheless, we have conducted the first investigation of the ISWI complex in *C. neoformans*, and there remain several unanswered questions and intriguing phenomena that require more examination and analysis. The study establishes a connection between post-translational regulation and antifungal drug resistance in *Cryptococcus neoformans*, thereby uncovering a previously unknown aspect of the emergence of drug resistance. We contend that the absence of *IOC* genes in the genome of *C. neoformans* does not yield a significant conceptual breakthrough in establishing definitive proof of the connection between Isw1 PTM, its function, and drug resistance. While we concur with the reviewer's opinion that studying the components and mechanisms of Isw1 regulation offers valuable insights into how *C. neoformans* responds to anti-fungal treatment and is relevant for clinical therapy, our future analysis aims to delve deeper into this intricate regulatory process of Isw1. We have made significant revisions to the manuscript, namely altering the description of what is considered "canonical". Please see lines 256-259.

**Author response image 4. sa2fig4:** 

2. It is unclear why the authors chose to focus on Chd1 and Sua7 as (1) neither Chd1 nor Sua7 appear in the AP-MS data and (2) there is very little evidence that Chd1 or Sua7 directly interact with Isw1 or Isw2 in *S. cerevisiae* (to the best of my knowledge, Michaelis et al., 2023 Nature provides the only biochemical evidence for interaction between Isw1 and Chd1 in *S. cerevisiae*). There is evidence that both Chd1 and Sua7 genetically interact with Isw1 or Isw2 in *S. cerevisiae*, which is what is described in the references cited by the authors (for example, Smolle et al., 2012). Along these lines, the authors should review the text and references and revise both accordingly. It should be noted that the evidence for an association between Isw1 and Chd1 in C. neoformans is quite interesting (although it may not be a direct physical interaction).

We really appreciate your feedback and sincerely apologize for erroneously identifying Chd1 and Sua7 as the interacting proteins with Isw1 from the literatures. After taking into account these crucial remarks, we have thoroughly examined the reference. There is no empirical data indicating that Chd1 and Sua7 have a protein-protein interaction with Isw1. In the study conducted by Smolle et al. in 2012, it was found that Chd1 and Isw1 had similar transcriptional controls on specific genes in the Set2 pathway. Consequently, this presented a compelling rationale and facet to investigate whether the simultaneous regulation of drug resistance occurs by *C. neoformans* Chd1 and Isw1. Our findings suggest that the control of drug resistance in *C. neoformans* is mediated by the Isw1-Itc1 complex pathway, rather than the Isw1-Chd1 pathway. Based on the Sua7 assays, there is currently no definitive evidence regarding the interaction. However, it has been observed that Sua7 mutants can influence the DNA binding targets of Isw2. It is therefore our primary intention to conduct experiments on these proteins. We concur with the reviewer's comments that these interactions reveal intriguing facets of Isw1's control. We intend to thoroughly examine them in the ongoing laboratory analysis to gather more detailed data for future analysis. The text has been altered, please see lines 135-139, 241-250, 471-474, and 479-482.

3. As mentioned in the initial comment, the authors are encouraged to review recent literature on ISWI homologs in organisms other than *S. cerevisiae* (e.g., *Neurospora crassa* – Kamei et al., 2021 PNAS and Wiles et al., 2022 eLife) to strengthen this aspect of the manuscript. For example, there are at least two additional proteins in the AP-MS data that are well-characterized to interact with ISWI homologs. Along these lines, the authors should provide protein names in addition to locus IDs in Figure 2a (as was done for Itc1) to help readers and broaden interest.

We concur with the comment. We have enhanced the introduction section by include additional details regarding ISWI complexes derived from various organisms. The modification of Figure 2a now includes gene names, so enhancing the provision of specific information to readers, please see lines 115-139, 241-242 and 1264.

Prior reviewer comment: The authors' use of "increase", "induce", or similar wording is confusing in some instances. For example, in Lines 197-198 the authors suggest that deacetylase inhibitors "increase acetylation levels" (see also Lines 235, 283, and 284 for additional examples). The authors should consider rewording these examples and reviewing the manuscript for similar examples to clarify their results.Author response: The manuscript has been appropriately updated in accordance with the given instructions.Reviewer response: Many such instances were not corrected (for example, see lines 281 and 322-323). Similarly, (1) in lines 127-128, the authors state "Cryptococcal Isw1 plays an indispensable role in modulating drug-resistance genes", but the role of Isw1 is not indispensable since the isw1 mutant strain has increased resistance to antifungals; and (2) in line 247, the authors state "that the isw1 mutant strain exhibits improved fungal burdens", but the isw1 mutant strain showed increased fungal burdens not improved fungal burdens.

We appreciate your thoughtful evaluations and comments. We agree that the aforementioned terms are inadequate in characterizing the data. They have been adjusted accordingly. Please see lines 309, 350-351, 153 and 275-276.